# SINGLE-HEAD ATTENTION IN HIGH DIMENSIONS: A THEORY OF GENERALIZATION, WEIGHTS SPECTRA, AND SCALING LAWS

## ABSTRACT

Trained attention layers exhibit striking and reproducible spectral structure of the weights, including low-rank collapse, bulk deformation, and isolated spectral outliers, yet the origin of these phenomena and their implications for generalization remain poorly understood. We study empirical risk minimization in a single-head tied-attention layer trained on synthetic high-dimensional sequence tasks generated from the attention-indexed model. Using tools from random matrix theory, spin-glass theory, and approximate message passing, we obtain an exact high-dimensional characterization of training and test error, interpolation and recovery thresholds, and the spectrum of the key and query matrices. Our theory predicts the full singular-value distribution of the trained query–key map—including low-rank structure and isolated spectral outliers—in qualitative agreement with observations in more realistic transformers. Finally, for targets with power-law spectra, we show that learning proceeds through sequential spectral recovery, leading to the emergence of power-law scaling laws.

## 1 INTRODUCTION

Attention mechanisms are the backbone of modern ML (Vaswani et al., 2017), yet their theoretical foundations remain elusive. Notably, trained attention layers exhibit non-random spectral patterns (Martin et al., 2021) and "stepwise" emergence during training (Wei et al., 2022). This motivates our central inquiry: *Can one propose and analytically solve a simplified model of attention in the high-dimensional regime that allows to explore the interplay between spectral properties and generalization?*

Our main contributions are
(i) **Asymptotics and inductive bias.** We derive exact formulas for training/test errors and recovery thresholds. We show that gradient-based algorithms converge to these solutions and demonstrate that factorized training (query/key matrices) provides a superior inductive bias over direct training of their product.
(ii) **Spectrum–generalization Link.** We provide an analytic characterization of the learned weight spectrum—predicting structured bulks, low-rank collapse, and outliers. This theory quantitatively links the learned spectrum to generalization performance, aligning with empirical transformer observations.
(iii) **Emergence and scaling Laws.** In regimes with power-law target spectra, we show that spectral modes are recovered sequentially. This isolates how target structure translates into sharp emergence phenomena and observable power-law scaling in attention layers.

## 2 SETTINGS

**Task and architecture.** We consider a sequence-to-sequence (seq2seq) and a sequence-to-label (seq2lab) supervised learning task. Our analyses applies to both these cases. In the seq2seq task we consider a dataset $\mathcal{D} = \{\boldsymbol{x}_{\text{in}}^{\mu}, \boldsymbol{x}_{\text{out}}^{\mu}\}_{\mu=1}^{n}$ where both the input and output data $\boldsymbol{x}_{\text{in}}^{\mu}, \boldsymbol{x}_{\text{out}}^{\mu} \in \mathbb{R}^{T \times d}$ are sequences of $T$ tokens, each given through an embedding vector $\boldsymbol{x}_a^{\mu} \in \mathbb{R}^d$ for $1 \leq a \leq T$. In the seq2lab task, instead we consider a dataset $\mathcal{D} = \{\boldsymbol{x}_{\text{in}}^{\mu}, y^{\mu}\}_{\mu=1}^{n}$ where the output data $y^{\mu} \in \mathbb{R}^{T \times T}$ are matrices of $T \times T$ pair-wise tokens comparisons.

In both cases, we aim at learning the input-output relationship through a parametrized function $\hat{f}(\boldsymbol{x}_{\text{in}}; W)$ of the form (respectively for the two tasks)

$$
\begin{cases}
\hat{f}_{\text{sq}}(\boldsymbol{x}_{\text{in}}; W) = A_W(\boldsymbol{x}_{\text{in}})\boldsymbol{x}_{\text{in}} \\
\hat{f}_{\text{lb}}(\boldsymbol{x}_{\text{in}}; W) = A_W(\boldsymbol{x}_{\text{in}})
\end{cases}
, \quad \text{where} \quad A_W(\boldsymbol{x}_{\text{in}}) = \sigma_\beta\left(\frac{\boldsymbol{x}_{\text{in}} W W^T \boldsymbol{x}_{\text{in}}^T - \mathbb{E}_{\text{tr}}[\boldsymbol{x} W W^T \boldsymbol{x}^T]}{d\sqrt{p}}\right)
\tag{1}
$$

with weights $W \in \mathbb{R}^{d \times p}$. $\sigma_\beta$ is the row-wise softmax activation at inverse temperature $\beta > 0$, and $A_W(\boldsymbol{x})$ is the attention matrix with tied key and query matrices. Thus, in the seq2seq task, $\hat{f}$ is a tied attention layer with identity value matrix. In the seq2lab case, instead, $\hat{f}$ outputs directly the attention matrix. Finally, $\mathbb{E}_{\text{tr}}$ is the empirical average over $\boldsymbol{x}$ in the training set, and the corresponding term plays the role of a batch-centering, ensuring a mean-zero input to the activation.

We learn $W$ by *empirical risk minimization* of the square loss with $\ell_2$ regularization (or equivalently, weight decay) that is commonly used in practice in large language models, i.e. $\hat{W} = \arg\min_W \mathcal{L}(W) = \arg\min_W [\mathcal{L}^{\text{data}}(W) + \mathcal{R}(W)]$ where respectively for the two tasks

$$
\begin{cases}
\mathcal{L}^{\text{data}}_{\text{sq}}(W) := \frac{1}{d} \sum_{\mu=1}^n \|\boldsymbol{x}^\mu_{\text{out}} - \hat{f}_{\text{sq}}(\boldsymbol{x}^\mu_{\text{in}}; W)\|_F^2, \\
\mathcal{L}^{\text{data}}_{\text{lb}}(W) := \sum_{\mu=1}^n \|y^\mu - \hat{f}_{\text{lb}}(\boldsymbol{x}^\mu_{\text{in}}; W)\|_F^2,
\end{cases}
\tag{2}
$$

and $\mathcal{R}(W) = \lambda\|W\|_F^2$. We measure the performance of the learned function $\hat{f}$ through the test errors

$$
e_{\text{test}}(\hat{f}) = \frac{1}{d} \mathbb{E}_{\boldsymbol{x}_{\text{in}}, \boldsymbol{x}_{\text{out}}} \|\boldsymbol{x}_{\text{out}} - \hat{f}(\boldsymbol{x}_{\text{in}})\|_F^2 \quad \text{or} \quad e_{\text{test}}(\hat{f}) = \mathbb{E}_{\boldsymbol{x}_{\text{in}}, y} \|y - \hat{f}(\boldsymbol{x}_{\text{in}})\|_F^2,
\tag{3}
$$

where $\mathbb{E}$ stands for an average over an appropriate test set. We remark that our analytical framework can be extended to a larger class of losses and regularization, see Section 3.

**Input data model.** To allow for analytical tractability, we assume that the input sequences $\boldsymbol{x}^\mu_{\text{in}}$ are Gaussian, i.e., that each token is independently given by $\boldsymbol{x}^\mu_{\text{in},a} \sim \mathcal{N}(0, \mathbb{I}_d)$ for $1 \le a \le T$. In this case, it can be shown that the centering term in Eq. (1) concentrates to $\mathbb{I}_T \text{Tr}(W W^T)$ in the regime $n \gg 1$. We remark that the high-dimensional setting we consider exhibits universality way beyond Gaussian data, in the same spirit as in Assumption 2.2 of Xu et al. (2025) and as proven by a number of recent works, e.g. Montanari & Saeed (2022); Dudeja et al. (2023); Dandi et al. (2023); Lu & Yau (2025).

**Target function model.** Following the successes of the single-index models, we consider a target function that lies within the expressivity class of the architecture in Eq. (1), and restrict the class of (possibly noisy) target functions to (respectively for the seq2seq and seq2lab tasks)

$$
\boldsymbol{x}^\mu_{\text{out}} = \sigma_{\beta_0}\left(R(\boldsymbol{x}_{\text{in}})\right)\boldsymbol{x}_{\text{in}} \quad \text{and} \quad y^\mu = \sigma_{\beta_0}\left(R(\boldsymbol{x}_{\text{in}})\right),
\tag{4}
$$

where $R(\boldsymbol{x}_{\text{in}}) \in \mathbb{R}^{T \times T}$ is a centered pre-activation matrix, and $\beta_0$ a softmax temperature possibly different from the learner's one $\beta$. We consider this model as we want to focus on the learnability of the token-to-token correlations exhibited by the output sequences. Concretely, we choose:

$$
R(\boldsymbol{x}_{\text{in}})_{ab} = \frac{\boldsymbol{x}_{\text{in,a}}^T S_0 \boldsymbol{x}_{\text{in,b}} - \delta_{ab} \text{Tr}(S_0)}{\sqrt{d}} + \sqrt{\frac{\Delta}{2 - \delta_{ab}}} \xi^\mu,
\tag{5}
$$

where $S_0$ is the target function weight matrix of rank $p_0$. We denoted $\xi^\mu \in \mathbb{R}^{T \times T}$ to be a symmetric standard Gaussian noise $\xi_{ab} = \xi_{ba} \sim N(0, 1)$ for all $1 \le a \le b \le T$, and the normalization of the variance ensures that the signal-to-noise ratio is uniform across tokens. The intuition behind Eq. (5) is that the learning model is structurally limited to expressing models of attention that are bilinear in the input sequences: for this reason, we model all higher-order dependencies as (Gaussian) noise $\xi$. For the test set, we consider samples distributed with the same distribution as the training set.

## 3 ASYMPTOTICS OF TRAIN AND TEST ERROR

Our main technical result is the characterization of the properties of the global minima of the empirical loss of Eq. (2) in the high-dimensional limit where the token embedding dimension $d \to \infty$ with quadratic number of samples and proportional ranks, i.e., the joint limit $d, n, p \to +\infty$ with

$$
\alpha := n/d^2, \quad \kappa := p/d, \quad \kappa_0 := p_0/d
\tag{6}
$$

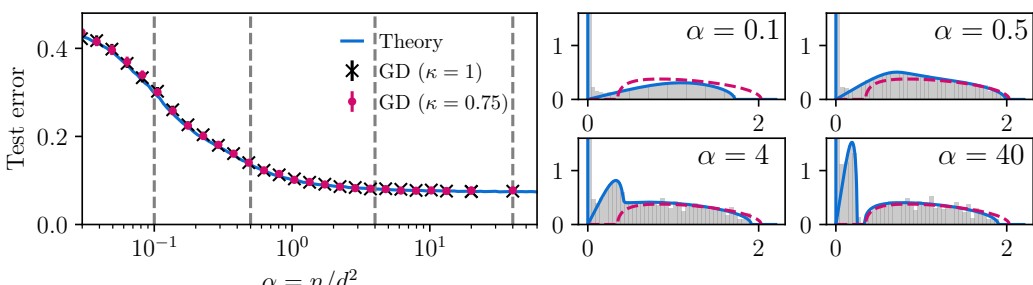

Figure 1: (Left) Test error of the ERM estimator Eq. (2) (Claim B.1, Eq. (14)) compared with Adam simulations at $d = 100$ (64 instances, error bars = standard deviation) as a function of the number of samples $\alpha = n/d^2$, where we use $\kappa = 0.75, 1$ (model width) and parameters $\lambda = 0.01$, $\Delta = 0.5$, $T = 2$, $\beta = \beta_0 = 1$ for the MP target (Section 3, $\kappa_0 = 0.5$). Theory and simulations agree. (Right) Singular value spectrum of the trained weights from theory (blue, Claim 4.1, Eq. (7)) vs. Adam simulations (grey histograms) at $\alpha = 0.05, 0.5, 4, 40$ ($d = 200$, 64 runs, 2000 samples in the test set). The asymptotic spectrum of the target is shown in red dashed. The theory also captures the delta peak at zero. For large $\alpha$, the spectrum splits into two bulks.

are $\mathcal{O}(1)$ ratios. We further assume that in the same limit the empirical spectral density of $S_0$ (which may be either random or deterministic) converges to a limiting distribution $\mu_0$ with a finite first and second moment. The length of the sequence $T = \mathcal{O}(1)$ is considered finite in our work in line with other works on sequence multi-index or attention-indexed models (Cui et al., 2024; Troiani et al., 2025; Boncoraglio et al., 2025). Our first result is Claim B.1, which predicts the train and test error of the global minimum of (2).

To illustrate our results, we consider the *Marchenko-Pastur (MP) target* case, where the weights $S_0$ are such that the limiting distribution satisfies $\mu_0(x) = \sqrt{\kappa_0}\mu_{\text{M.P.}}(\sqrt{\kappa_0}x)$, where $\mu_{\text{M.P.}}$ is the Marchenko-Pastur distribution (Marchenko & Pastur, 1967) with normalized rank $0 < \kappa_0 < 1$, and $Q_0 = 1 + \kappa_0$. For example, this is the case in which the target attention matrix has the form $S_0 = W_0 W_0^T / \sqrt{p_0 d}$, where $W_0 \in \mathbb{R}^{d \times m_0}$ has i.i.d. components extracted from a distribution with zero mean and unit variance, and $\kappa_0 = p_0/d = \text{rank}(S_0)/d$. We believe that the overall phenomenology we showcase here is valid qualitatively for generic $S_0$ with $\text{rank}(S_0) = \kappa_0 d$ and positive eigenvalues bounded away from the origin. Additionally, we consider the case of matched softmax temperatures $\beta = \beta_0 = 1$ (noticing that a mismatch in temperatures can always be reabsorbed in a rescaling of the regularization). In Figure 1, we plot the test error predicted by our theory as a function of the sample ratio $\alpha = n/d^2$ for $\Delta = 0.5$, which perfectly matched the experiments of gradient descent.

See Appendix B for the formula of the asymptotics and further discussions.

## 4 SPECTRUM AND GENERALIZATION

Our second result is the characterization of the spectrum of the ERM estimator (2) and its implication for generalization.

**Claim 4.1** (Spectra of ERM). *Under the same assumptions and notations of Claim B.1, let $\hat{W}$ be a global minimizer of ERM in the high-dimensional limit. Then, in distribution,*

$$\frac{1}{\sqrt{pd}}\hat{W}^T\hat{W} \stackrel{d}{=} \eta\,\text{ReLU}(S_0 + \delta Z - \epsilon\,I_d)\,, \tag{7}$$

*where $Z \sim \text{GOE}(d)$, the $\text{ReLU}$ is applied on the spectrum, and we defined $\eta = \hat{m}^\star/\hat{\Sigma}^\star$, $\delta = \sqrt{\hat{q}^\star}/\hat{m}^\star$, $\epsilon = 2\tilde{\lambda}/\hat{m}^\star$. In particular, the empirical spectral density of $\hat{W}^T\hat{W}/\sqrt{pd}$ converges for $d \to \infty$ to a deterministic limit fully determined by $(\eta, \delta, \epsilon)$ and the spectral law of $S_0$.*

A key consequence of Claim B.1 is that both the learned spectrum and the test error are governed by the scalar parameters $(\eta, \delta, \epsilon)$ arising from the minimization of (9). This yields an explicit link between generalization and spectral structure: $\delta$ controls the effective noise level due to finite training set size, while $\epsilon$ controls the regularization strength, setting small eigenvalues to zero and inducing low effective rank and spectral outliers.

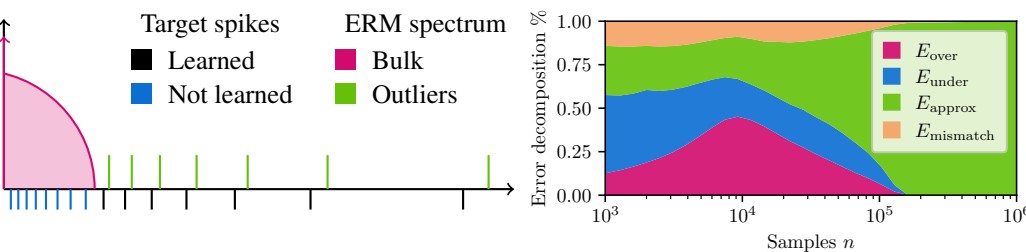

Figure 2: Qualitative representation of the spectral density and error decomposition of Section 4. (Top) Spectrum of the target (below the horizontal axis) and of the ERM (above the axis). The ERM spectrum is composed of outliers (learned target features) and a noise bulk (magenta). (Bottom) Error decomposition for power-law target $\gamma = 0.75$, $d = 200$, $\Delta = 0.5$, $T = 2$, $\lambda = 1/d$ (see Appendix J), expressed as fraction of the total error (shown in Figure 4).

**Spectral law for low-rank targets.** We start by showcasing the resulting spectral behavior in the case of MP target in the quadratic sample regime (6). Figure 1 compares the spectrum of the learned weights with that of target weights (as well as with numerical simulations with gradient descent). We observe an agreement between theory and experiments, and we clearly see that for this target function learning happens through a sequence of qualitative behaviors (rank-collapse, bleed-out, appearance of outliers) aligning more and more with the target distribution as the number of samples increases. This is in qualitative agreement with spectra observed experimentally in real architectures (Martin et al., 2021; Martin & Mahoney, 2021; Staats et al., 2024).

**From spectrum to generalization.** We make the connection between spectral properties and generalization explicit by studying how the test error approaches its limiting values as the number of samples increases. To do so, we need to restrict our attention to targets with eigenvalues $\{s_i\}_{i=1}^{p}$, which we interpret as a set of learnable features that decay faster than $i^{-1/2}$ (including sub-linear rank targets $p_0 = o(d)$).

Assuming further that the learned weights are well aligned with the target (small test error), the learned spectrum admits a simple and interpretable form (see sketch in Figure 2). It consists of a semicircular bulk, originating from noise and finite-sample effects, together with a finite number of outlying eigenvalues that are in one-to-one correspondence with the recovered target eigenvalues $\{s_i\}$, up to a regularization-dependent rescaling and thresholding: spectral outliers can be interpreted as learned features, while the bulk captures spurious directions induced by noise. Up to the leading order, the test error at the empirical minimum can then be written

$$e_{\text{test}}(\hat{W}) = e_0 + e_1(n, d, \lambda) \quad \text{with} \quad e_1 = c_1\big(E_{\text{over}} + E_{\text{under}} + E_{\text{approx}}\big) + c_2 E_{\text{mismatch}}, \quad (8)$$

where $e_0$ is a saturation value independent of $n, d, \lambda$, and the decay term $e_1$ depends explicitly on the learned spectrum. $c_1, c_2$ depend only on the activations and the label noise. This excess error decomposes into four contributions:

(a) $E_{\text{over}}$ is an *overfitting term*, controlled by the noisy bulk of the learned spectrum and arising from fitting spurious directions induced by finite samples;

(b) $E_{\text{under}}$ is an *underfitting term*, corresponding to target directions that are not recovered due to noise or regularization, consisting of the eigenvalues still trapped in the bulk.

(c) $E_{\text{approx}}$ is an *approximation term*, measuring residual error along recovered spectral outliers, i.e. learned features;

(d) Finally, $E_{\text{mismatch}}$ is a *mismatch term*, which appears when the saturation value $e_0$ does not correspond to a local minimum of the generalization error.

Explicit expressions for these terms are given in Appendix H. Taken together, this decomposition shows that spectral outliers correspond to genuinely learned target features, and that increasing the number of recovered outliers systematically improves generalization. Conversely, the bulk of small eigenvalues represents pure noise arising from finite samples: the learned spectrum provides a compact and interpretable summary of both the inductive bias induced by the regularization and its consequences for generalization.

As an application of the error decomposition, we can derive exact learning curves and universal exponents for power-law targets. See Appendix C.

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

## A  RELATED WORK

Modern machine learning relies increasingly on attention mechanisms Vaswani et al. (2017), which form the backbone of state-of-the-art models in natural language processing, vision, and beyond Devlin et al. (2019); Dosovitskiy et al. (2020); Brown et al. (2020). Despite their empirical success, many fundamental questions about learning with attention layers remain open.

In particular, the weight matrices of trained attention layers display striking and reproducible spectral patterns. Empirical studies report that the singular-value distributions of the query and key projections are far from random: they exhibit a structured bulk with non-trivial tails Martin et al. (2021); Staats et al. (2024), consistent with earlier observations of heavy-tailed and compressible spectra across deep networks Mahoney & Martin (2019). These regularities persist across architectures and training scales, yet their theoretical origin and their implications for generalization remain largely unexplained. *Why do such spectra emerge in attention weights, and what do they reveal about the inductive biases of the model?*

Second, neural scaling laws Kaplan et al. (2020); Brown et al. (2020); Hoffmann et al. (2022) reveal striking power-law relations between performance and computational or data resources in attention-based models. Yet, with few exceptions, theoretical understanding of these phenomena remains largely confined to linearized or lazy regimes, e.g. Caponnetto & De Vito (2007); Cui et al. (2021); Maloney et al. (2022); Bahri et al. (2024); Paquette et al. (2024); Atanasov et al. (2024); Bordelon et al. (2024); Kunstner & Bach (2025); Defilippis et al. (2024) that may not capture genuine representation learning. At the same time, growing evidence suggests that learning in attention layers is not smooth: training dynamics often display plateaus and abrupt transitions, with new directions in representation space emerging sequentially rather than all at once. This stepwise behavior has been observed both in simplified attention models and in large-scale transformers Wei et al. (2022); Raventós et al. (2023); Arora & Goyal (2023); Schaeffer et al. (2023); Ren et al. (2025). *Can emergence and scaling phenomena of this kind be reproduced and explained within a solvable attention model?*

A natural strategy for addressing these questions is to analyze simplified models in high-dimensional regimes, where the blessing of dimensionality Donoho et al. (2000) enables sharp and tractable characterizations of learning. This approach has driven major advances in the theory of two-layer networks via Gaussian single- and multi-index models (see, e.g., Ben Arous et al. (2021); Abbe et al. (2022); Ba et al. (2022); Arnaboldi et al. (2023); Damian et al. (2023); Bietti et al. (2025); Berthier et al. (2024)). In these settings, high-dimensional asymptotics have enabled an essentially complete characterization of empirical risk minimization.

In contrast, attention mechanisms remain far less explored within such a high-dimensional framework (with notable exceptions of low-rank attention Cui et al. (2024); Troiani et al. (2025); Arnaboldi et al. (2025) and in-context linear regression Zhang et al. (2025)), and, to our knowledge, there is no high-dimensional asymptotic theory linking spectral properties to generalization.

Several empirical studies have further noted that attention projections are approximately low-rank. Staats et al. (2024) found rapidly decaying spectra in pretrained transformers, while Si et al. (2025) reported similar profiles in LLaMA models, showing that the low-rank tendency is present already at pretraining and persists through fine-tuning. Broader surveys confirm stable low-rank structure during training Yunis et al. (2024), and parameter-efficient fine-tuning methods such as LoRA Hu et al. (2022) exploit the fact that very low ranks suffice to capture relevant structure. Recent work further leverages spectral profiles for efficient adaptation Si et al. (2025). While these findings are empirical, recent theoretical progress has linked weight decay on the query and key matrices to an effective nuclear-norm regularization of their product Kobayashi et al. (2024), thereby formalizing

a low-rank inductive bias in attention, but they do not characterize any explicit relation between this low-rank bias and generalization. The low-rank inductive bias connects directly to earlier analyses of factorized parameterizations in feedforward networks, where weight decay implicitly promotes low-rank solutions through nuclear-norm penalties Gunasekar et al. (2017; 2018); Arora et al. (2019); Galanti et al. (2024).

In contrast to the extensive literature on single- and multi-index models, attention mechanisms remain comparatively underexplored from a high-dimensional theoretical perspective. Existing analyses of attention Cui et al. (2024); Troiani et al. (2025); Arnaboldi et al. (2025) primarily focus on extremely low-rank regimes of order $\mathcal{O}(1)$, which do not capture the rich spectral structure observed empirically in trained attention layers. More recently, attention-indexed models have been introduced Boncoraglio et al. (2025). These models offer a controlled setting in which questions of generalization and representation learning can be addressed. However, existing results in this line focus on Bayes-optimal inference, leaving open the characterization of solutions reached by empirical risk minimization (ERM), which is the regime relevant for trained neural networks.

Our work builds on recent progress in the asymptotic analysis of quadratic neural networks Erba et al. (2025); Xu et al. (2025); Defilippis et al. (2025) to bridge this gap by mapping ERM in attention to an equivalent matrix sensing problem. This reformulation allows us to analyze the learned spectrum using tools from approximate message passing and its connections to convex and non-convex optimization Berthier et al. (2020); Gerbelot & Berthier (2023); Loureiro et al. (2021); Vilucchio et al. (2025).

## B  TECHNICAL RESULTS CONCERNING THE ASYMPTOTICS

**Claim B.1** (Train and test error of ERM)**.** *Define $\tilde{\lambda} = \sqrt{\kappa}\lambda$ and $\mu_\delta = \mu_0 \boxplus \mu_{\mathrm{sc},\delta}$, where $\boxplus$ is the free convolution and $\mu_{\mathrm{sc},\delta}(x) = \sqrt{4\delta^2 - x^2}/(2\pi\delta^2)$ the semicircle distribution of radius $2\delta$ for $\delta > 0$, and $F_\delta$ its c.d.f.. Define also $\tilde{\sigma}_\beta(A) = \sigma_\beta(\{\sqrt{1 + \delta_{ab}}\, A_{ab}\}_{ab})$ for $A \in \mathbb{R}^{T \times T}$, and where $\sigma_\beta$ is applied row-wise. Call $(\Sigma^\star, m^\star, q^\star, \hat{\Sigma}^\star, \hat{m}^\star, \hat{q}^\star)$ the global minimizer of*

$$\Phi(\Sigma, m, q, \hat{\Sigma}, \hat{m}, \hat{q}) = \frac{\hat{q}\Sigma + 2\hat{m}m - \hat{\Sigma}q}{4} + \frac{n}{d^2}\mathcal{M}(\Sigma, m, q) - \frac{\hat{m}^2}{4\hat{\Sigma}}J\left(\frac{\sqrt{\hat{q}}}{\hat{m}}, \frac{2\tilde{\lambda}}{\hat{m}}\right), \qquad (9)$$

*where*

$$\mathcal{M}(\Sigma, m, q) = \mathbb{E}_{z_0, z} \inf_{h \in \mathcal{S}(T)} \left\{ \frac{1}{2\Sigma} \sum_{\substack{a,b=1 \\ a \leq b}}^{T} (h_{ab} - z_{ab})^2 + \sum_{a,b=1}^{T} [\tilde{\sigma}_{\beta_0}(z_0)_{ab} - \tilde{\sigma}_\beta(h)_{ab}]^2 \right\},$$

$$(10)$$

$$J(\delta, \epsilon) = \int_\epsilon^{+\infty} \mu_\delta(\mathrm{d}x)\,(x - \epsilon)^2,$$

*$\mathcal{S}(T)$ is the set of $T \times T$ symmetric matrices, and where $\mathbb{E}_{z_0, z}$ denotes the average over $\{z_{ab}, z_{0,ab}\}_{1 \leq a \leq b \leq T}$, which are Gaussian random variables*

$$\begin{bmatrix} z_{0,ab} \\ z_{ab} \end{bmatrix} \sim \mathcal{N}\left( \begin{bmatrix} 0 \\ 0 \end{bmatrix}; \begin{bmatrix} Q_0 + \Delta/2 & m \\ m & q \end{bmatrix} \right), \qquad (11)$$

*independently for each pair $a, b$ (up to symmetry $z_{ab} = z_{ba}$ and $z_{0,ab} = z_{0,ba}$). Assume that at the global minimum the replicon condition Vilucchio et al. (2025) is satisfied:*

$$2\alpha \mathbb{E}_{z_0, z} \sum_{\substack{a,b,c,d=1 \\ a \leq b,\, c \leq d}}^{T} \left( \frac{\partial_{z_{ab}} p_{cd} - \delta_{ac}\delta_{bd}}{\Sigma} \right)^2 \int \mu_{\sqrt{\hat{q}}/\hat{m}}(\mathrm{d}x)\, \mu_{\sqrt{\hat{q}}/\hat{m}}(\mathrm{d}y)\, \frac{(\xi(x) - \xi(y))^2}{\hat{\Sigma}^2(x - y)^2} < 1, \qquad (12)$$

*where $\xi(x) = ReLU(x - 2\tilde{\lambda}/\hat{m})$ and*

$$p(z_0, z, \Sigma) = \underset{h \in \mathcal{S}(T)}{\operatorname{arginf}} \left\{ \frac{1}{2\Sigma} \sum_{\substack{a,b=1 \\ a \leq b}}^{T} (h_{ab} - z_{ab})^2 + \sum_{a,b=1}^{T} [\tilde{\sigma}_{\beta_0}(z_0)_{ab} - \tilde{\sigma}_\beta(h)_{ab}]^2 \right\}. \qquad (13)$$

*Then, for all values of $\alpha, \lambda > 0$, $\Delta \geq 0$ and $\kappa \geq 1 - F_{\sqrt{\hat{q}}/\hat{m}}(2\tilde{\lambda}/\hat{m})$ any global minimum $\hat{W}$ of Eq.* (2) *satisfies*

$$\lim_{d \to \infty} \mathbb{E}e_{\text{test}}(\hat{W}) = \mathbb{E}_{z_0,z} \sum_{a,b=1}^{T} [\tilde{\sigma}_{\beta_0}(z_0)_{ab} - \tilde{\sigma}_{\beta}(z)_{ab}]^2 \, , \tag{14}$$

$$\lim_{d \to \infty} d^{-2} \mathbb{E}\mathcal{L}(\hat{W}) = \Phi(\Sigma, m, q, \hat{\Sigma}, \hat{m}, \hat{q}) \, ,$$

*where all order parameters $(\Sigma, m, q, \hat{\Sigma}, \hat{m}, \hat{q})$ are evaluated at the global minimum defined above.*

Claim B.1 characterizes the global minimum of (2) through the solution of a six-dimensional optimization problem (9), whose numerical solution we discuss in Appendix J. To provide a bit more intuition, we remark the parameters $m^\star, q^\star$ and $\Sigma^\star$ measure, respectively, the target-learner overlap $\text{Tr}(S_0^\top S)/d$, the learner norm $\text{Tr}(S^\top S)/d$ and the trace of the inverse Hessian of the loss Clarte et al. (2023) as customary in approximate-message-passing analysis, where we defined $S = \hat{W}\hat{W}^\top/\sqrt{pd}$. We also stress that Claim B.1 can be adapted to more general losses $\sum_{\mu=1}^{n} \ell\left(\boldsymbol{x}_\mu S_0 \boldsymbol{x}_\mu^T; \boldsymbol{x}_\mu S \boldsymbol{x}_\mu^T\right)$ (i.e., depending bi-linearly on the data) and regularizations inducing a spectral penalty on the effective matrix $S = W^T W/\sqrt{pd}$. This includes models with mismatched nonlinearities between data and learner, classification tasks, and more. We provide the equations for the more general case in Appendix E.2.1.

We formulate the result as a claim rather than a theorem, as its complete proof would require a substantial technical development that closely follows standard arguments from high-dimensional inference. We present a proof sketch highlighting the key steps in Appendix E. All components of the argument rely on established techniques from random matrix theory and approximate message passing. Additionally, we stress that we evaluated condition (12) numerically for all cases shown in the plots, and we found that it was always satisfied.

**Sharp learning curves at quadratic sample complexity.** In Figure 1 (and Figure 6 in the Appendix J) we plot the test error predicted by our theory as a function of the sample ratio $\alpha = n/d^2$ for $\Delta = 0.5$. We observe a monotone decreasing behavior at large values of regularization $\lambda$ (Figure 1), while at lower values of $\lambda$ a characteristic interpolation peak (with oddly asymmetric shape) appears around the maximal number of samples that can be perfectly fit by the model architecture, the so-called interpolation threshold (Figure 6). Claim B.1 allows to derive an analytic prediction for the interpolation threshold, i.e. the value $\alpha_{\text{interp}}$ before which the ERM estimator for $\lambda \to 0^+$ achieves zero training loss. In the noiseless case $\Delta = 0$, we can also predict analytically the perfect recovery threshold (Corollary B.2), i.e. the value $\alpha_{\text{perf}}$ after which the ERM estimator achieves perfect generalization. B.1.

**Corollary B.2** (Interpolation and perfect recovery thresholds)**.** *Consider the setting of Section 2 with $\lambda \to 0^+$, $\Delta \geq 0$ and $T \geq 2$. Then, there exists a value of sample ratio $\alpha_{\text{interp}}$ such that the training loss at its global minimum is zero for $\alpha < \alpha_{\text{interp}}$ (perfect fit of the training set), and strictly positive for $\alpha > \alpha_{\text{interp}}$, and $\alpha_{\text{interp}}$ satisfies*

$$\alpha_{\text{interp}} = \frac{\partial_1 J\left(\bar{\delta}, 0\right)}{2\bar{\delta}(T^2 + T - 2)} \tag{15}$$

*where $\bar{\delta}$ is the solution of*

$$Q_0 + \frac{\Delta}{2} = J\left(\bar{\delta}, 0\right) - \frac{\bar{\delta}}{2}\partial_1 J\left(\bar{\delta}, 0\right) \tag{16}$$

*and $J$ is defined in Eq.* (10)*. Moreover, if $\Delta = 0$, there exists a value of sample ratio $\alpha_{\text{perf}}$ such that the test error at the global minimum is zero for $\alpha > \alpha_{\text{perf}}$ (perfect generalization), and strictly positive for $\alpha < \alpha_{\text{perf}}$, determined as follows. Call $\bar{c}$ the solution of the equation*

$$M_{\text{s.c.}}^{(1)}(c) - c M_{\text{s.c.}}^{(0)}(c) + \frac{c}{1 - \kappa_0} = 0, \tag{17}$$

*where*

$$M_{\text{s.c.}}^{(k)}(x) = \int_{-2}^{x} dx\, \mu_{\text{sc}}(x)\, x^k \tag{18}$$

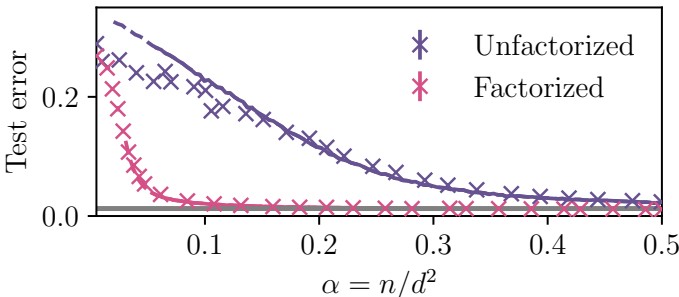

Figure 3: Test error as a function of $\alpha$ for standard factorized query–key training Eq. (2) and direct (non-factorized) training of the attention matrix Eq. (21), both at optimal regularization (selected by cross-validation). The factorized parameterization consistently achieves significantly lower test error across all $\alpha$. Solid lines show theoretical predictions (Claim B.1 and Appendix G), while dots correspond to numerical experiments with Adam at $d = 100$, averaged over 8 runs with 2000 test samples. Parameters are $\kappa_0 = 0.05$, $\Delta = 0.05$, $T = 2$, and $\beta = \beta_0 = 1$.

for $0 < \kappa_0 < 1$. *Then,*

$$\alpha_{\mathrm{perf}} = \frac{1 - (1 - \kappa_0)^2 \left( M_{\mathrm{s.c.}}^{(2)}(c) - c M_{\mathrm{s.c.}}^{(1)}(c) \right)}{T^2 + T - 2} \,. \tag{19}$$

Corollary B.2 follows from a mapping of the theory of softmax attention to that of linear attention that holds for $\alpha \leq \alpha_{\mathrm{interp}}$ and $\lambda \to 0^+$, allowing to adapt results from Erba et al. (2025), as detailed in Appendix F.

**Behavior of gradient descent.** A natural question is whether gradient-based methods can reach global minima of the non-convex loss in Eq. (2), or whether they remain stuck in spurious local minima. In Fig. 1 and 6 we compare our theory with numerical results of runs of Adam Kingma & Ba (2015) (details in Appendix J) run on the loss of Eq. (2) with $d = 100$ and averaging over 32 different instances, and $p = d$. We observe an excellent match for both the test error and the training loss for both $\kappa = 1$ and $\kappa = 3/4$.

**Inductive bias: weight decay implies nuclear norm on attention.** The $\ell_2$ regularization over the weights $W$ naturally translates to a nuclear norm regularization in the equivalent generalized matrix problem, see Kobayashi et al. (2024) and Appendix D, naturally favoring model weights configurations with an effective lower width (i.e., implementable with fewer hidden units): a weight decay in $W$ thus implies a low-rank learned attention matrix. This partly explains why learning with low-rank weights (usually done for computational reasons) does not negatively affect generalization despite reducing expressivity. We quantify this bias in Claim B.1, where we find that all attention learners with rank satisfying $p/d \geq 1 - F_{\sqrt{\bar{q}}/\hat{m}}(2\tilde{\lambda}/\hat{m})$ achieve the same error. This implies that mildly rank-constrained to massively over-parametrized architectures incur no penalty in terms of train and test error.

**Comparison with non-factorized parameterization.** A natural baseline to consider is learning a non-factorized attention $S$ through the empirical risk minimization of the model (e.g. in the seq2seq task)

$$\hat{f}_{\mathrm{L2}}(\boldsymbol{x}_{\mathrm{in}}; S) = \sigma_\beta \left( \frac{\boldsymbol{x}_{\mathrm{in}} S \boldsymbol{x}_{\mathrm{in}}^T - \mathbb{E}_{\mathrm{tr}} \operatorname{Tr}(\boldsymbol{x}_{\mathrm{in}} S \boldsymbol{x}_{\mathrm{in}}^T)}{\sqrt{d}} \right) \boldsymbol{x}_{\mathrm{in}} \,, \tag{20}$$

where we impose that $S$ is symmetric (but neither PSD nor factorized), and we train on the loss

$$\mathcal{L}_{\mathrm{L2}}(S) := \frac{1}{d} \sum_{\mu=1}^{n} ||\boldsymbol{x}_{\mathrm{out}}^\mu - \hat{f}_{\mathrm{L2}}(\boldsymbol{x}_{\mathrm{in}}^\mu; S)||_2^F + \lambda ||S||_F^2 \,. \tag{21}$$

The Frobenius-regularized estimator underperforms the nuclear-norm–regularized one whenever the target is sufficiently low-rank. In the noisy case, we compare the test error of the factorized ERM

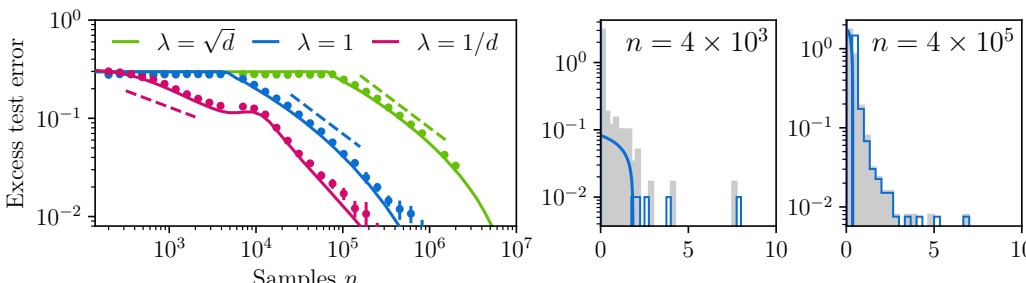

Figure 4: (Left) Excess test error (Eq. (3) minus its value at $n \to +\infty$) of the ERM estimator Eq. (2) (Claim B.1, Eq. (14)) compared with Adam simulations at $d = 200$ (8 instances, error bars = standard deviation) as a function of sample number $n$, where we use power-law target (Section C, decay exponent $\gamma = 0.75$), and $\Delta = 0.5$, $T = 2$, $\beta = \beta_0 = 1$. We plot three values of $\lambda = 1/d, 1, \sqrt{d}$. One can see clearly several scaling regimes in different regions, with different decay exponents. We highlight those of them (in dashed lines) in which the error decays as $n^{-f(\gamma)}$ with non-trivial dependency on $\gamma$, namely $f(\gamma) = 1 - 1/(2\gamma)$ for $\lambda = 1/d$, and $f(\gamma) = 2 - 1/\gamma$ for $\lambda = 1, \sqrt{d}$. (Right) Eigenvalue spectrum of the ERM estimator from theory (blue, Claim 4.1) vs. Adam simulations (grey histograms) at $n = 4 \cdot 10^3, 4 \cdot 10^5$ ($d = 400$, single runs). For large $n$ the spectrum develops a heavy tail.

estimator of Eq. (2) with the non-factorized Frobenius estimator of Eq. (21) at optimal regularization (cross-validated, Fig. 3 left). The factorized model consistently outperforms, fully exploiting the induced low-rank bias. In the extreme low-rank regime $p_0 \ll d$, it achieves vanishing error at sample scale $\mathcal{O}(dp_0)$, whereas the non-factorized model requires $\mathcal{O}(d^2)$, see Fig. 8 left in Appendix J.

## C  POWER-LAW TARGETS AND SCALING LAWS

We now focus on targets whose attention map exhibits power-law structure, as discussed for multi-index models in Ren et al. (2025); Ben Arous et al. (2025); Defilippis et al. (2025). Concretely, we assume that the target matrix $S_0$ has a power-law spectrum, with eigenvalues $\{\sqrt{d}i^{-\gamma}\}_{i=1}^{d}$ with $\gamma > 1/2$, corresponding to a sequence of signal directions with progressively decreasing strength. Such spectra are natural in structured or multi-scale tasks and provide a minimal setting in which emergence and scaling phenomena can be studied analytically. All details of the derivations are reported in Appendix I.

**Sequential spectral recovery and heavy-tailed spectra.** Under the power-law assumption, the learned spectral law (7) features a progressive recovery of signal modes. The effective threshold $\epsilon$ acts as a moving cutoff along the target spectrum: as the sample complexity $\alpha$ increases (or the regularization decreases), $\epsilon$ shifts toward smaller eigenvalues, allowing increasingly weaker spectral directions of $S_0$ to be incorporated into the learned representation. Learning, therefore, proceeds through a sequence of regimes where a growing fraction of the dominant spectral modes contributes to generalisation.

This mechanism extends the spectral phenomenology discussed earlier. For targets with Marchenko–Pastur–type spectra, the same process leads to the emergence of a bulk of spectral outliers as learning progresses. Instead, when the target attention map has a power-law spectrum the progressive recovery of spectral modes produces isolated outliers, reproducing one of the most robust empirical observations in trained attention layers (Figure 4 and Martin et al. (2021); Staats et al. (2024)), and leads in some regimes to a learned spectrum with power-law tails. The heavy-tailed behavior arises not from isolated spikes, but from the accumulation of an increasing number of recovered modes across scales.

Specifying our results to power-law targets also leads to an interpretable mechanism for emergence Wei et al. (2022); Schaeffer et al. (2023); Ren et al. (2025). The underfitting and approximation terms of (8) are controlled by the cutoff scale $\epsilon$, while the overfitting contribution is governed by the bulk noise level $\delta$. As $n, d, \lambda$ varies, different contributions dominate the excess risk, leading to

sharp crossovers between learning regimes. These crossovers correspond to the successive inclusion of new spectral modes into the learned attention map, providing a first-principles explanation of emergence phenomena in feature-learning models. From this perspective, the diversity of spectral behaviors observed in modern attention architectures would reflect differences in task structure, rather than qualitative changes in optimization dynamics or architectural design.

**Power-law learning curves and universal exponents.** The same emergence mechanism directly yields power-law scaling of the test error with the number of samples. In particular, the excess risk obeys $e_{\text{test}} - e_0 \asymp n^{-f(\gamma)}$ where the exponent $f(\gamma) > 0$ depends on the tail exponent $\gamma$ of the target spectrum, and on the value of $n, d, \lambda$. Remarkably, these rates coincide with those previously obtained for LASSO and matrix compressed sensing by Defilippis et al. (2025), supporting their universality, and read:

$$e_{\text{test}} = e_0 + \Theta\left( \left(\frac{n}{d}\right)^{-1+\frac{1}{2\gamma}} + \left(\frac{n}{d^2}\right)^{\frac{2}{5}} \right) \tag{22}$$

for $d \ll n \ll d^2$, $\lambda \ll \sqrt{\frac{n}{d^2}}$ and

$$e_{\text{test}} - e_0' = \begin{cases} \Theta\left(\frac{d^2}{n}\right), & n \gg d^2, \ \lambda \ll \sqrt{\frac{n}{d^2}}, \\ \Theta\left( \left(\frac{\lambda d^{3/2}}{n}\right)^{2-\frac{1}{\gamma}} \right), & \max\left(\sqrt{\frac{n}{d^2}}, \frac{n}{d^{\gamma+3/2}}\right) \ll \lambda \ll \frac{n}{d}, \\ \Theta\left( \left(\frac{\lambda d^2}{n}\right)^2 \right), & \sqrt{\frac{n}{d^2}} \ll \lambda \ll \frac{n}{d^{\gamma+3/2}} . \end{cases} \tag{23}$$

where $e_0$ and $e_0' \le e_0$ refer to the two saturation errors. The derivation is detailed in Appendix I, and the results are illustrated in Fig. 4 as well as in a phase diagram in Fig. 5 in Appendix I. The key mechanism underlying these universal scaling laws is thus the combination of two ingredients: *quasi-sparsity* of the target representation, encoded by a heavy-tailed spectrum that induces an ordered hierarchy of feature strengths Mallat (1999); Donoho (2006), and the implicit *rank-sparsity* bias induced by attention. Under optimal regularization, these yield an excess generalization error scaling as $n^{-1+1/(2\gamma)}$, matching the classical minimax rate achieved by the LASSO Raskutti et al. (2011).

# D    INDUCTIVE NUCLEAR NORM BIAS FOR TIED AND UNTIED WEIGHTS

We briefly recall here the mathematical reasons behind the appearance of the *inductive nuclear norm*, following arguments that go back to classical variational characterizations of atomic norms, and were emphasized in the deep learning context by Neyshabur et al. (2015); Gunasekar et al. (2017); Soudry et al. (2018); Pesme & Flammarion (2023). These arguments are quite general: whenever a predictor is parameterized in a factorized form (for instance, as the product of two matrices, or as a diagonal interaction of two vectors), an $\ell_2$-type penalty on the factors induces, via an AM–GM–style variational identity, an $\ell_1$ or nuclear norm penalty on the effective predictor. More recent works Maillard & Kunisky (2024); Xu et al. (2025) have also used these ideas.

**Case 1 (Tied weight):**   Let $W \in \mathbb{R}^{d \times p}$ and $M := W^\top W \in \mathbb{R}^{p \times p}$. Assume the loss depends on $W$ only via $M$, i.e. there exists $\Phi : \mathbb{S}_+^p \to \mathbb{R}$ such that

$$\mathcal{L}(W) = \Phi(W^\top W) + \lambda \|W\|_F^2.$$

Then the following problems are equivalent in optimal value, and their minimizers correspond via $M = W^\top W$:

$$\min_{W \in \mathbb{R}^{d \times p}} \Phi(W^\top W) + \lambda \|W\|_F^2 \quad \equiv \quad \min_{M \succeq 0, \ \text{rank}(M) \le d} \Phi(M) + \lambda \|M\|_\star. \tag{$\star$}$$

*Proof.* Since $M = W^\top W \succeq 0$ with $\text{rank}(M) \le d$, and if $s_i(W)$ are the singular values of $W$, then the eigenvalues of $M$ are $s_i(W)^2$. Hence

$$\|W\|_F^2 = \sum_i s_i(W)^2 = \sum_i \lambda_i(M) = \|M\|_\star.$$

Because $\Phi$ depends only on $M$, we can replace $W$ by $M$ and obtain ($\star$). Moreover, any feasible $M \succeq 0$ with $\text{rank}(M) \le d$ admits a factorization $M = W^\top W$; choosing such a $W$ makes the two objectives equal. $\qquad\square$

**Case 2 (Untied weights):** Let $U \in \mathbb{R}^{p \times r}$, $V \in \mathbb{R}^{n \times r}$, and $M := UV^\top \in \mathbb{R}^{p \times n}$. Assume the loss depends on $(U, V)$ only through the product $M$, i.e.

$$\mathcal{L}(U, V) = \Psi(UV^\top) + \frac{\lambda}{2} \big( \|U\|_F^2 + \|V\|_F^2 \big),$$

for some (possibly nonconvex) function $\Psi : \mathbb{R}^{p \times n} \to \mathbb{R}$. We show the exact equivalence

$$\min_{U,V} \Psi(UV^\top) + \tfrac{\lambda}{2}(\|U\|_F^2 + \|V\|_F^2) \equiv \min_{M \in \mathbb{R}^{m \times n}} \Psi(M) + \lambda \|M\|_\star. \tag{†}$$

**Matrix AM–GM analogue (variational identity for $\|\cdot\|_*$).** The key inequality is the matrix analogue of AM–GM: for any factorization $M = UV^\top$,

$$\|M\|_\star \leq \tfrac{1}{2} \big( \|U\|_F^2 + \|V\|_F^2 \big), \tag{A}$$

with equality attained by a specific SVD-based factorization.

*Proof of (A).* Use the dual characterization of the nuclear norm:

$$\|M\|_\star = \max_{\|A\|_2 \leq 1} \langle M, A \rangle = \max_{\|A\|_2 \leq 1} \langle UV^\top, A \rangle = \max_{\|A\|_2 \leq 1} \langle U, AV \rangle,$$

where $\langle X, Y \rangle := \mathrm{Tr}(X^\top Y)$ and $\|\cdot\|_2$ is the spectral norm. For any $A$ with $\|A\|_2 \leq 1$, apply Cauchy–Schwarz and the scalar AM–GM (or $ab \leq \frac{1}{2}(a^2 + b^2)$):

$$\langle U, AV \rangle \leq \|U\|_F \|AV\|_F \leq \|U\|_F \|A\|_2 \|V\|_F \leq \|U\|_F \|V\|_F \leq \tfrac{1}{2} \big( \|U\|_F^2 + \|V\|_F^2 \big).$$

Taking the maximum over $\|A\|_2 \leq 1$ yields (A).

**Tightness of (A).** Let the compact SVD of $M$ be $M = P\Sigma Q^\top$ with $\Sigma = \mathrm{diag}(\sigma_1, \ldots, \sigma_\rho)$, $\sigma_i \geq 0$. Choose the *balanced SVD factorization*

$$U := P\Sigma^{1/2}, \qquad V := Q\Sigma^{1/2}.$$

Then $M = UV^\top$ and

$$\|U\|_F^2 + \|V\|_F^2 = \mathrm{Tr}(\Sigma) + \mathrm{Tr}(\Sigma) = 2 \sum_{i=1}^{\rho} \sigma_i = 2\|M\|_\star.$$

Hence, equality holds in (A):

$$\|M\|_\star = \min_{M = UV^\top} \tfrac{1}{2} \big( \|U\|_F^2 + \|V\|_F^2 \big), \tag{B}$$

and the minimum is attained by the balanced SVD factorization.

**Equivalence of objectives.** Because $\Psi$ depends on $(U, V)$ only through $M = UV^\top$, we may fix $M$ and minimize the regularizer over all factorizations $M = UV^\top$. By (B),

$$\inf_{U,V : UV^\top = M} \frac{\lambda}{2} \big( \|U\|_F^2 + \|V\|_F^2 \big) = \lambda \|M\|_\star.$$

Therefore

$$\inf_{U,V} \Psi(UV^\top) + \tfrac{\lambda}{2}(\|U\|_F^2 + \|V\|_F^2) = \inf_M \Psi(M) + \lambda \|M\|_\star,$$

which is exactly (†).

Notice that no convexity of $\Psi$ is required for the equality of optimal values; convexity only affects algorithmic guarantees.

# E   DERIVATION OF CLAIM B.1 AND PROOF SKETCH

Note that Claim 1 requires evaluating the replicon condition (12), which guarantees convergence of the AMP algorithm Bolthausen (2014). Although a general proof is difficult and direct evaluation can be numerically demanding, in our experiments AMP did converge (the replicon condition being a convergence criterion for AMP), and gradient-based algorithms reached the same fixed point. This provides strong empirical evidence that the replicon condition is satisfied in our setting.

In this paper, we provide an informal justification for Claim B.1. Claim B.1 can be proven by going through the following steps:

- Reduction to the PSD generalized matrix estimation problem

$$\tilde{\tilde{\mathcal{L}}}(S) := \frac{1}{d}\sum_{\mu=1}^{n}||\tilde{\sigma}_\beta(\{\mathrm{Tr}[SX_{ab}(\boldsymbol{x}^\mu)]\}_{ab})\boldsymbol{x}^\mu - \tilde{\sigma}_{\beta_0}(\{\mathrm{Tr}[S_0 X_{ab}(\boldsymbol{x}^\mu)]\}_{ab}))\boldsymbol{x}^\mu||_F^2 + \sqrt{pd}\,\lambda\,\mathrm{Tr}(S) \tag{24}$$

  where $X_{ab}(\boldsymbol{x}) = (\boldsymbol{x}_a\boldsymbol{x}_b^T + \boldsymbol{x}_b\boldsymbol{x}_a^T - 2\delta_{ab}\mathbb{I}_d)/\sqrt{2d(1+\delta_{ab})}$, $S = W^T W/\sqrt{pd} \succeq 0$ and $\tilde{\sigma}(A) = \sigma(\{\sqrt{1+\delta_{ab}}A_{ab}\}_{ab})$. This step is discussed in Appendix D.

- Asymptotic equivalence of sequence-to-sequence and sequence-to-attention tasks. One needs to show (as done for e.g. in Cui et al. (2024)) that Eq. (24) is asymptotically the same as

$$\tilde{\mathcal{L}}(S) := \sum_{\mu=1}^{n}||\tilde{\sigma}_\beta(\{\mathrm{Tr}[SX_{ab}(\boldsymbol{x}^\mu)]\}_{ab}) - \tilde{\sigma}_{\beta_0}(\{\mathrm{Tr}[S_0 X_{ab}(\boldsymbol{x}^\mu)]\}_{ab})||_F^2 + d\tilde{\lambda}\,\mathrm{Tr}(S)\,, \tag{25}$$

  where we called $\tilde{\lambda} = \sqrt{p/d}\lambda = \sqrt{\kappa}\lambda$. To do this, one should use a law-of-large-numbers kind of argument giving

$$\frac{1}{d}||F(\boldsymbol{x})\boldsymbol{x}||_F^2 = \frac{1}{d}F(\boldsymbol{x})\boldsymbol{x}\boldsymbol{x}^T F(\boldsymbol{x})^T \approx F(\boldsymbol{x})F(\boldsymbol{x})^T \tag{26}$$

  where we used that $\mathbb{E}_{\boldsymbol{x}}\boldsymbol{x}\boldsymbol{x}^T/d = \mathbb{I}_T$.

- Gaussian universality, i.e. replacement of each matrix $X_{ab}(\boldsymbol{x}^\mu)$ by a random independent Wigner matrix $G_{ab}^\mu$. This step is *a priori* non-trivial, and its proof would require generalizing the arguments of Maillard et al. (2024); Xu et al. (2025); Erba et al. (2025) to the case of multiple tokens (the case $T = 1$ is instead included in Erba et al. (2025)). We do not foresee any technical roadblock here, as this step requires only promoting scalar outputs to finite-dimensional vectorial ones in all such proofs.

- Analysis of the final empirical risk minimization problem

$$\tilde{\mathcal{L}}(S) := \frac{1}{d}\sum_{\mu=1}^{n}||\sigma_\beta(\{\mathrm{Tr}[SG_{ab}^\mu]\}_{ab}) - \sigma_{\beta_0}(\{\mathrm{Tr}[S_0 G_{ab}^\mu]\}_{ab})||_F^2 + d\tilde{\lambda}\,\mathrm{Tr}(S)\,. \tag{27}$$

  The asymptotic analysis of this loss can be done by adapting the analysis in Erba et al. (2025) (corresponding to the single token case $T = 1$ with linear activation) to multiple tokens, in the same spirit as the generalization of single-index models to multiple tokens performed in Cui et al. (2024). In practice this reduces to:

  - Writing down an appropriate Approximate Message Passing (AMP) algorithm (which we present explicitly in the following) whose fixed point are local minima of the loss in Eq. (27).
  - Using AMP theory to derive a set of low-dimensional state evolution equations that track AMP step-by-step, and use them to characterize the properties of its fixed points.
  - Among the fixed points of state evolution, select the one with the lowest value of the training loss (describing then the properties of global minima).

  The only non-trivial point here is that this program is *a priori* that for non-convex losses such as Eq. (27) AMP may not converge signaling the onset of so-called replica symmetry breaking. Nonetheless, in Vilucchio et al. (2025) the authors rigorously show that the described procedure is correct whenever the replicon condition in Eq. (12) is satisfied.

Again one would need to generalize their proof (an AMP upper bound coupled with a lower-bound in the line of Stojnic (2009); Thrampoulidis et al. (2014; 2020)) to multiple tokens, but that should not pose any roadblock.

We devote the rest of the section to write down the suitable AMP algorithm used in the sketch of the proof, in the more generic setting of multi-token generalized matrix sensing.

### E.1 GENERIC SETTING

Consider the data model

$$y^\mu = g_0(\{\mathrm{Tr}(S_0 Z_a^\mu)\}_{a=1}^{L_{\mathrm{in}}})\,, \tag{28}$$

where $Z_a^\mu$ are GOE$(d)$ matrices, $S_0 \in \mathbb{R}^{d \times d}$ and $g_0 : \mathbb{R}^{L_{\mathrm{in}}} \to \mathbb{R}^{L_{\mathrm{out}}}$. $g_0$ can be a stochastic function, in which case we assume that its stochasticity is independent and identical for each sample $\mu$. Consider the empirical risk minimization problem over $d \times d$ symmetric matrices

$$\hat{S} = \arg\min_{S \in \mathcal{C}} L(S)\,, \qquad L(S) = \sum_\mu \ell\left(y^\mu; \{\mathrm{Tr}(S Z_a^\mu)\}_{a=1}^{L_{\mathrm{in}}}\right) + R(S)\,, \tag{29}$$

for $\ell : \mathbb{R}^{L_{\mathrm{out}}} \times \mathbb{R}^{L_{\mathrm{in}}} \to \mathbb{R}$, $R : \mathbb{R}^{d \times d} \to \mathbb{R}$ a rotationally invariant regularization (i.e. $R(S) = R(OSO^T)$ for all $d$-dimensional rotation matrices $O$) and $\mathcal{C}$ is a rotationally-invariant subset of the set of $d \times d$ symmetric matrices.

**Special case I: main text model.** In the main text, we consider $L_{\mathrm{in}} = T(T+1)/2$ (interpreted as $T \times T$ symmetric matrices), $L_{\mathrm{out}} = T^2$, $g_0$ is the row-wise softmax with Gaussian noise in the input

$$g_0(A) = \sigma\left(\left\{\frac{A_{ab} + \sqrt{\Delta}\xi_{ab}}{\sqrt{2 - \delta_{ab}}}\right\}_{ab}\right) \tag{30}$$

where $\xi^\mu \in \mathbb{R}^{T \times T}$ is a symmetric standard Gaussian noise $\xi_{ab} = \xi_{ba} \sim N(0,1)$ for all $1 \le a \le b \le T$,

$$\ell(y, \hat{y}) = \sum_{a,b=1}^T (y_{ab} - \hat{y}_{ab})^2 \quad \text{and} \quad R(S) = d\tilde{\lambda}\,\mathrm{Tr}(S)\,. \tag{31}$$

Finally, $\mathcal{C}$ is the set of PSD symmetric $d \times d$ matrices.

**Special case II: main text model with linear attention.** In this case, we consider $L_{\mathrm{in}} = T(T+1)/2$ and $L_{\mathrm{out}} = T(T+1)/2$ (interpreted as $T \times T$ symmetric matrices), $g_0$ is

$$g_0(A)_{ab} = \frac{A_{ab} + \sqrt{\Delta}\xi_{ab}}{\sqrt{2 - \delta_{ab}}}\,, \tag{32}$$

where $\xi^\mu \in \mathbb{R}^{T \times T}$ is a symmetric standard Gaussian noise $\xi_{ab} = \xi_{ba} \sim N(0,1)$ for all $1 \le a \le b \le T$, and

$$\ell(y, \hat{y}) = \sum_{a,b=1}^T (y_{ab} - \hat{y}_{ab})^2 \quad \text{and} \quad R(S) = d\tilde{\lambda}\,\mathrm{Tr}(S)\,. \tag{33}$$

Finally, $\mathcal{C}$ is the set of PSD symmetric $d \times d$ matrices. The case of single token $T = 1$ reduces to Erba et al. (2025).

### E.2 MAPPING TO A VECTOR-WEIGHTS MODEL WITH COUPLED REGULARIZATION

We consider the mapping from vec : $\mathrm{Sym}_d \to \mathbb{R}^{d(d+1)/2}$ (which conveniently maps the Frobenius scalar product in $\mathrm{Sym}_d$ given by $\langle A; B \rangle = \mathrm{Tr}(AB)$ to the standard Euclidean scalar product in $\mathbb{R}^{d(d+1)/2}$) given by

$$\mathrm{vec}\,(A)_{(ab)} = \left\langle b^{(ab)}; A \right\rangle = \sqrt{2 - \delta_{ab}}A_{ab}\,, \tag{34}$$

under the choice of orthonormal basis

$$b_{ij}^{(aa)} = \delta_{ia}\delta_{ja}\,, \quad b_{ij}^{(ab)} = \frac{\delta_{ia}\delta_{jb} + \delta_{ib}\delta_{ja}}{\sqrt{2}}\,. \tag{35}$$

Here $(ab)$ stands for the ordered pair of $1 \leq a \leq b \leq d$, and we denote $A_{ij}$ as the $i, j$ entry of a matrix $A$, while as $A_{(ab)}$ the component of matrix $A$ onto the basis element $b^{(ab)}$. Let us denote $d(d+1)/2 = D$ (we use $D \approx d^2/2$ as we are interested in the leading order in $d$).

Our input data is given, for each sample $\mu$, by $L_{\text{in}}$ $d \times d$ symmetric matrices $\{Z_a\}_{a=1}^{L_{\text{in}}}$ that in the Gaussian equivalent model we treat as independent GOEs. Then the sensing vectors satisfy

$$A_{a,(ij)} = \sqrt{\frac{d}{2}}\text{vec}\,(Z_a)_{(ij)} \sim N(0, \mathbb{I}_D) \tag{36}$$

for $1 \leq a \leq L_{\text{in}}$, and $(ij)$ means the dimension indices $1 \leq i \leq j \leq d$. Moreover, if we define

$$w_{(ij)} = \sqrt{\frac{d}{2}}\text{vec}\,(S)_{(ij)}, \tag{37}$$

we have that

$$\text{Tr}(SZ_a) = \sum_{(ij)} Z_{a,(ij)}S_{(ij)} = \sqrt{\frac{2}{d}} \sum_{(ij)} A_{a,(ij)}S_{(ij)} = \frac{\sqrt{2}}{\sqrt{D}} \sum_{(ij)} A_{a,(ij)}w_{(ij)},$$
$$\frac{1}{d}\text{Tr}(S^2) = \frac{1}{d}\sum_{(ij)} S_{(ij)}^2 = \frac{2}{d^2}\sum_{(ij)} w_{(ij)}^2 = \frac{1}{D}\sum_{(ij)} w_{(ij)}^2. \tag{38}$$

In this formulation, under Gaussian equivalence, we see that our model is just a sequence single-index model of the form Cui et al. (2024) with coupled prior term.

Finally, the loss in the vectorial model reads

$$\tilde{\ell}(y = g_0(z_0), z) = \ell(y = g_0(\sqrt{2}z_0), \sqrt{2}z) \tag{39}$$

due to the factor $\sqrt{2}$ in Eq. (38).

### E.2.1 Approximate Message Passing

Under the mapping of Section E.2 we can directly combine the single token derivation of AMP given in Erba et al. (2025) (which holds for generic single-token losses and generic rotational-invariant regularization), with multi-token AMP given in Cui et al. (2024) to obtain the following AMP algorithm with fixed points given by local minima of the loss in Eq. (29). We stress that the multi-token extension can be mapped to the standard treatment of AMP, amounting to a non-separable non-linearity along the samples dimension. Hence the mapping from ERM to AMP of Erba et al. (2025) can be adapted to the multi-token case.

**AMP in vector notation.** In vector notation, the labels are generated as

$$y = g_0(\{\sqrt{2}D^{-1/2} \sum_{(ij)} A_{\mu,a,(ij)}^0 w_{(ij)}\}_{a=1}^{L_{\text{in}}}) \tag{40}$$

and the AMP reads (here $\mu = 1, \dots, n$, $a = 1, \dots, L_{\text{in}}$ and $1 \leq i \leq j \leq d$)

$$w_{(ij)}^t = \sqrt{\frac{d}{2}}\text{vec}\left(\phi(\sqrt{\frac{2}{d}}\text{mat}\,(\Gamma^{t-1}), \Lambda^{t-1})\right)_{(ij)}$$

$$V^t = \frac{1}{D}(\text{div}\phi)(\sqrt{\frac{2}{d}}\text{mat}\,(\Gamma^{t-1}), \Lambda^{t-1})$$

$$\omega_{\mu,a}^t = \frac{1}{\sqrt{D}}\sum_{(ij)} A_{\mu,a,(ij)}w_{(ij)}^t - \theta(t \geq 1)V^t f_{\mu,a}^{t-1}$$

$$f_{\mu,a}^t = \frac{\text{prox}(y_\mu, \omega_\mu^t, V^t)_a - \omega_{\mu,a}^t}{V} \tag{41}$$

$$\Lambda^t = -\frac{1}{D}\sum_{a=1}^{L_{\text{in}}}\sum_{\mu=1}^{n} \partial_{\omega_{\mu,a}} f_{\mu,a}^t$$

$$\Gamma_{(ij)}^t = \frac{1}{\sqrt{D}}\sum_{a=1}^{L_{\text{in}}}\sum_{\mu=1}^{n} A_{\mu,a,(ij)} f_{\mu,a}^t + \Lambda^t w_{(ij)}^t$$

where

$$\text{prox}(y, \omega, V) = \underset{h \in \mathbb{R}^{L_{\text{in}}}}{\text{arginf}} \left\{ \frac{1}{2V} \sum_{a=1}^{L_{\text{in}}} (h_a - \omega_a)^2 + \ell(y, \sqrt{2}h) \right\} \tag{42}$$

$$\phi(M = ODO^T, \Lambda) = O \underset{T \in \mathcal{C}}{\arg\min} \left[ \frac{1}{d^2} R(T) + \frac{\Lambda}{4} \sum_{i=1}^{d} T_i^2 - \frac{1}{2} \sum_{i=1}^{d} T_i D_i \right] O^T$$

where $M$ is a $d$-dimensional symmetric matrix with eigen-decomposition $M = ODO^T$.

**AMP in matrix notation.** If we map back all quantities to their original matrix shape, we obtain the following equivalent AMP algorithm. The labels are generated as

$$y^\mu = g_0(\{\text{Tr}(S_0 Z_a^\mu)\}_{a=1}^{L_{\text{in}}}) \tag{43}$$

and the AMP reads (here $\mu = 1, \ldots, n$, $a = 1, \ldots, L_{\text{in}}$ and $1 \leq i, j \leq d$)

$$S^t = \phi(\Theta^{t-1}, \Lambda^{t-1})$$

$$V^t = \frac{2}{d^2} (\text{div}\phi)(\Theta^{t-1}, \Lambda^{t-1})$$

$$\Omega_{\mu,a}^t = \text{Tr}(S^t Z_a^\mu) - \theta(t \geq 1) V^t F_{\mu,a}^{t-1}$$

$$F_{\mu,a}^t = \frac{\text{prox}_{\text{mat}}(y^\mu, \Omega_\mu^t, V^t)_a - \Omega_{\mu,a}^t}{V} \tag{44}$$

$$\Lambda^t = -\frac{2}{d^2} \sum_{a=1}^{L_{\text{in}}} \sum_{\mu=1}^{n} \partial_{\Omega_{\mu,a}} F_{\mu,a}^t$$

$$\Theta_{ij}^t = \frac{1}{d} \sum_{a=1}^{L_{\text{in}}} \sum_{\mu=1}^{n} Z_{a,ij}^\mu F_{\mu,a}^t + \Lambda^t S_{ij}^t$$

where $\phi$ is the same as in the vector case, while

$$\text{prox}(y, \omega, V)_{\text{mat}} = \underset{h \in \mathbb{R}^{L_{\text{in}}}}{\text{arginf}} \left\{ \frac{1}{4V} \sum_{a=1}^{L_{\text{in}}} (h_a - \omega_a)^2 + \ell(y, h) \right\}. \tag{45}$$

**State evolution.** The iterations of both algorithms can be tracked by the following state evolution equations.

$$\begin{cases} \hat{q}^t & = \frac{2n}{d^2(\Sigma^t)^2} \mathbb{E}_{z_0,z} \sum_{a=1}^{L_{\text{in}}} (p_a - z_a)^2 \\ \hat{\Sigma}^t & = \frac{2n}{d^2\Sigma^t} \left[ L_{\text{in}} - \mathbb{E}_{z_0,z} \sum_{a=1}^{L_{\text{in}}} \frac{Q_0 z_a p_a - m(z_0)_a p_a}{Q_0 q - m^2} \right] \\ \hat{m}^t & = \frac{2n}{d^2\Sigma^t} \mathbb{E}_{z_0,z} \sum_{a=1}^{L_{\text{in}}} \frac{q(z_0)_a p_a - m z_a p_a}{Q_0 q - m^2} \\ m^{t+1} & = -\partial_{\hat{m}} \Psi(\hat{\Sigma}^t, \hat{q}^t, \hat{m}^t) \\ q^{t+1} & = 2\partial_{\hat{\Sigma}} \Psi(\hat{\Sigma}^t, \hat{q}^t, \hat{m}^t) \\ \Sigma^{t+1} & = -2\partial_{\hat{q}} \Psi(\hat{\Sigma}^t, \hat{q}^t, \hat{m}^t) \end{cases} \tag{46}$$

where

$$p_a = \underset{h \in \mathbb{R}^{L_{\text{in}}}}{\text{arginf}} \left\{ \frac{1}{2\Sigma^t} \sum_{a=1}^{L_{\text{in}}} (h_a - \omega_a)^2 + \ell(g_0(\sqrt{2}\{z_0\}), \sqrt{2}h) \right\}, \tag{47}$$

$$\Psi(\hat{\Sigma}, \hat{q}, \hat{m}) = \frac{2}{d} \mathbb{E}_D \min_{T \in \mathcal{C}} \left[ \frac{1}{d^2} R(T) + \frac{\hat{\Sigma}}{4} \sum_{i=1}^{d} T_i^2 - \frac{1}{2} \sum_{i=1}^{d} T_i D_i \right],$$

where the average $\mathbb{E}_D$ is over the spectrum $D$ of the matrix $\hat{m}S_0 + \sqrt{\hat{q}}Z$ with $Z \sim \text{GOE}(d)$, while $\mathbb{E}_{z_0,z}$ is over two Gaussian vectors $z_0, z \in \mathbb{R}^{L_{\text{in}}}$ such that, independently for all components $a = 1, \ldots, L_{\text{in}}$

$$\begin{bmatrix} (z_0)_a \\ z_a \end{bmatrix} \sim \mathcal{N} \left( \begin{bmatrix} 0 \\ 0 \end{bmatrix}; \begin{bmatrix} Q_0 & m^t \\ m^t & q^t \end{bmatrix} \right), \tag{48}$$

and over any stochasticity of the activation $g_0$.

**Observables.** Call $T^\star$ is the optimizer in $\Psi$ of Eq. (47). Then, by state evolution, at the global minimum $\hat{W}$ of Eq. (29), we have that the spectral density of $\hat{W}\hat{W}^T/\sqrt{pd}$ converges to the one of $T^\star$. Moreover, pre-activations converge to Gaussians of the form

$$\begin{bmatrix} \mathrm{Tr}(S_0 Z_a) \\ \mathrm{Tr}(S Z_a) \end{bmatrix} \sim \mathcal{N}\left( \begin{bmatrix} 0 \\ 0 \end{bmatrix}; \begin{bmatrix} 2Q_0 & 2m \\ 2m & 2q \end{bmatrix} \right) \tag{49}$$

when evaluated on data $Z$ that is not included in the training set. Otherwise, the target's pre-activations are still Gaussian with zero mean and variance $2Q_0$, but the learned model's pre-activations are given by

$$\mathrm{Tr}(S Z_a) \sim \sqrt{2} p_a \,. \tag{50}$$

With that in mind, we have that the training error (without regularization term) at a fixed point of AMP is given by

$$e_{\mathrm{train}} = \mathbb{E}_{z_0,z} \sum_{a=1}^{L_{\mathrm{in}}} \ell\left( g_0(\sqrt{2} z_0); \sqrt{2} p \right) \,, \tag{51}$$

The regularization part of the loss can be computed as $d^{-2} R(T^\star)$, and any test loss (intended as a comparison of the output of the target and learned function) can be instead computed as

$$e_{\mathrm{test}} = \mathbb{E}_{z_0,z} \sum_{a=1}^{L_{\mathrm{in}}} \ell_{\mathrm{test}}\left( g_0(\sqrt{2} z_0); \sqrt{2} z \right) \,. \tag{52}$$

All averages are as defined for the state evolution equations.

**Variational formulation.** It can be checked Cui et al. (2024) that Eq. (9) is stationary at the fixed point of Eq. (46), and that it matches the training loss.

**Replicon condition.** The replicon condition is the linear stability condition of AMP under perturbations of a fixed points. A derivation is given in Erba et al. (2025). In our case, we have

$$\frac{2\alpha}{\hat{\Sigma}^2 \Sigma^2} \mathbb{E}_{z_0,z} \sum_{\substack{a,b,c,d=1 \\ a\leq b,\, c\leq d}}^{T} (\partial_{z_{ab}} p_{cd} - \delta_{ac}\delta_{bd})^2 \frac{2}{d^2} \mathbb{E}_D \sum_{\substack{1\leq i\leq j\leq d \\ 1\leq k\leq l\leq d}} \partial_{M_{ij}} \phi(M = ODO^T, \hat{\Sigma}) < 1 \tag{53}$$

where the averages are the same as for state evolution.

### E.3 Simplifications in the main text setting

In the main text, we consider $L_{\mathrm{in}} = T(T+1)/2$ (interpreted as $T \times T$ symmetric matrices), $L_{\mathrm{out}} = T^2$, $g_0$ is the row-wise softmax with Gaussian noise in the input

$$g_0(A) = \sigma_{\beta_0}(\{[A_{ab} + \sqrt{\Delta}\xi_{ab}]/\sqrt{2 - \delta_{ab}}\}_{ab}) \,, \tag{54}$$

where $\xi \in \mathbb{R}^{T \times T}$ is a symmetric standard Gaussian noise $\xi_{ab} = \xi_{ba} \sim N(0,1)$ for all $1 \leq a \leq b \leq T$,

$$\ell(y, z) = \sum_{a,b=1}^{T} (y_{ab} - \sigma_\beta(z_{ab}/\sqrt{2 - \delta_{ab}}))^2 \quad \text{and} \quad R(S) = d\tilde{\lambda}\, \mathrm{Tr}(S) \,. \tag{55}$$

In this case,

$$\phi(M = ODO^T, \Lambda) = O \,\mathrm{diag}\left( \frac{\mathrm{ReLU}(D_{ii} - 2\tilde{\lambda})}{\Lambda} \Big| i = 1, \ldots, d \right) O^T$$

$$\Psi(\hat{\Sigma}, \hat{q}, \hat{m}) = -\frac{\hat{m}^2}{2\hat{\Sigma}} J(\sqrt{\hat{q}}/\hat{m}, 2\tilde{\lambda}/\hat{m}) \tag{56}$$

$$J(a, b) = \int_b^{+\infty} dx \, \mu_a(x) \, (x - b)^2 \,.$$

where $\mu_a = \mu_{\mathrm{sc}} \boxplus \mu_0$. Notice that the noise can be treated simply by altering the second moment of $z_0$ in the state evolution equations from $Q_0$ to $Q_0 \to Q_0 + \Delta/2$ (where the factor 2 comes from gathering the $\sqrt{2}$ factor present in the state evolution equations).

**Observables.** The training and test errors are given by

$$
e_{\text{train}} = \mathbb{E}_{z_0,z} \sum_{a=1}^{L_{\text{in}}} ||\tilde{\sigma}_{\beta_0}(\sqrt{2}z_0) - \tilde{\sigma}_\beta(\sqrt{2}p)||_F^2 \, ,
$$

$$
e_{\text{test}} = \mathbb{E}_{z_0,z} \sum_{a=1}^{L_{\text{in}}} ||\tilde{\sigma}_{\beta_0}(\sqrt{2}z_0) - \tilde{\sigma}_\beta(\sqrt{2}z)||_F^2 \, .
$$

$$(57)$$

The spectral density of $\hat{W}\hat{W}^T/\sqrt{pd}$ converges to that of $T^\star$, which is a shifted, rescaled and cropped version of the spectral density of $S_0 + \sqrt{\hat{q}}/\hat{m}Z$ for $Z \sim \text{GOE}(d)$, giving the expression in the main text.

**State evolution.** This gives the following state equations (at the fixed point)

$$
q = \frac{\hat{m}^2}{\hat{\Sigma}^2} J\left(\frac{\sqrt{\hat{q}}}{\hat{m}}, \frac{2\tilde{\lambda}}{\hat{m}}\right)
$$

$$
m = \frac{\hat{m}J\left(\frac{\sqrt{\hat{q}}}{\hat{m}}, \frac{2\tilde{\lambda}}{\hat{m}}\right) - \tilde{\lambda}\partial_2 J\left(\frac{\sqrt{\hat{q}}}{\hat{m}}, \frac{2\tilde{\lambda}}{\hat{m}}\right) - \frac{\sqrt{\hat{q}}}{2}\partial_1 J\left(\frac{\sqrt{\hat{q}}}{\hat{m}}, \frac{2\tilde{\lambda}}{\hat{m}}\right)}{\hat{\Sigma}}
$$

$$
\Sigma\hat{\Sigma} = \frac{\hat{m}}{2\sqrt{\hat{q}}}\partial_1 J\left(\frac{\sqrt{\hat{q}}}{\hat{m}}, \frac{2\tilde{\lambda}}{\hat{m}}\right)
$$

$$
\Sigma\hat{\Sigma} = 2\alpha L - 2\alpha\mathbb{E}_{z_0,z} \sum_{a=1}^{L_{\text{in}}} \frac{Q_0 z_a p_a - m z_{0,a} p_a}{Q_0 q - m^2}
$$

$$
\Sigma\hat{m} = 2\alpha\mathbb{E}_{z_0,z} \sum_{a=1}^{L_{\text{in}}} \frac{q z_{0,a} p_a - m z_a p_a}{Q_0 q - m^2}
$$

$$
\hat{q}\Sigma^2 = 2\alpha\mathbb{E}_{z_0,z} \sum_{a=1}^{L_{\text{in}}} (p_a - z_a)^2
$$

$$(58)$$

where $\alpha = n/d^2$ and

$$
p_a = \underset{h \in \mathbb{R}^{L_{\text{in}}}}{\text{arginf}} \left\{ \frac{1}{2\Sigma} \sum_{a=1}^{L_{\text{in}}} (h_a - \omega_a)^2 + \sum_{a=1}^{L_{\text{out}}} (\tilde{\sigma}_{\beta_0}(\sqrt{2}z_0)_a - \tilde{\sigma}_\beta(\sqrt{2}h)_a)^2 \right\} .
$$

$$(59)$$

For the replicon, the only thing that changes is that the regularization dependent part becomes the same as presented in Erba et al. (2025), giving the expression of Claim B.1.

**Small regularization limit $\lambda \to 0^+$.** To study the small regularization limit before interpolation (where a vanishing regularization would lead to degenerate global minima), one can perform the

change of variables $\hat{m} \to \hat{m}\tilde{\lambda}$, $\hat{q} \to \hat{q}\tilde{\lambda}^2$, $\Sigma \to \Sigma/\tilde{\lambda}$ and $\hat{\Sigma} \to \hat{\Sigma}\tilde{\lambda}$ to obtain

$$q = \frac{\hat{m}^2}{\hat{\Sigma}^2} J\left(\frac{\sqrt{\hat{q}}}{\hat{m}}, \frac{2}{\hat{m}}\right)$$

$$m = \frac{\hat{m}J\left(\frac{\sqrt{\hat{q}}}{\hat{m}}, \frac{2}{\hat{m}}\right) - \tilde{\lambda}\partial_2 J\left(\frac{\sqrt{\hat{q}}}{\hat{m}}, \frac{2}{\hat{m}}\right) - \frac{\sqrt{\hat{q}}}{2}\partial_1 J\left(\frac{\sqrt{\hat{q}}}{\hat{m}}, \frac{2}{\hat{m}}\right)}{\hat{\Sigma}}$$

$$\Sigma\hat{\Sigma} = \frac{\hat{m}}{2\sqrt{\hat{q}}}\partial_1 J\left(\frac{\sqrt{\hat{q}}}{\hat{m}}, \frac{2}{\hat{m}}\right)$$

$$\Sigma\hat{\Sigma} = 2\alpha L - 2\alpha\mathbb{E}_{z_0,z}\sum_{a=1}^{L_{\text{in}}} \frac{Q_0 z_a p_a - m z_{0,a} p_a}{Q_0 q - m^2} \qquad (60)$$

$$\Sigma\hat{m} = 2\alpha\mathbb{E}_{z_0,z}\sum_{a=1}^{L_{\text{in}}} \frac{q z_{0,a} p_a - m z_a p_a}{Q_0 q - m^2}$$

$$\hat{q}\Sigma^2 = 2\alpha\mathbb{E}_{z_0,z}\sum_{a=1}^{L_{\text{in}}} (p_a - z_a)^2$$

with

$$p_a = \operatorname*{arginf}_{h \in \mathbb{R}^{L_{\text{in}}}} \left\{ \frac{\tilde{\lambda}}{2\Sigma}\sum_{a=1}^{L_{\text{in}}} (h_a - \omega_a)^2 + \sum_{a=1}^{L_{\text{out}}} (\tilde{\sigma}_{\beta_0}(\sqrt{2}z_0)_a - \tilde{\sigma}_\beta(\sqrt{2}h)_a)^2 . \right\} \qquad (61)$$

We then see that for $\tilde{\lambda} \to 0^+$ the last expression reduces to

$$p_a = \operatorname*{arginf}_{h \in \mathbb{R}^{L_{\text{in}}}} \left\{ \sum_{a=1}^{L_{\text{out}}} (\tilde{\sigma}_{\beta_0}(\sqrt{2}z_0)_a - \tilde{\sigma}_\beta(\sqrt{2}h)_a)^2 \right\} . \qquad (62)$$

with the prescription that if such arginf is degenerate, one should pick the arginf closest to $z$ in L2 distance. This set of equations is valid as long as $\Sigma > 0$ and not diverging. At interpolation we expect $\Sigma \to 0$, as there the curvature of the loss (proportional to $\Sigma^{-1}$) diverges.

After interpolation, there is no need to change variable. One can put directly $\lambda \to 0^+$ in the original Eq. (46). This set of equations is valid as long as $\Sigma > 0$ and not diverging. At interpolation we expect $\Sigma \to +\infty$, as there the curvature of the loss (proportional to $\Sigma^{-1}$) goes to zero.

# F  REDUCTION TO LINEAR ATTENTION

Let us consider the setting of the main text in the $\lambda \to 0^+$ limit, before interpolation (i.e. when there exists multiple sets of weights that perfectly fit the training dataset). Then, we need to solve Eq. (68). We now show that here, for the case of softmax-softmax studied in the main text, the equations reduce to a rescaled version of the state equations for the case of linear single token attention Erba et al. (2025). We consider w.l.o.g. the case $\beta = \beta_0$. Let us consider the proximal

$$p_a = \operatorname*{argmin}_{h \in \mathbb{R}^{L_{\text{in}}}} \left\{ \sum_{a=1}^{L_{\text{out}}} (\tilde{\sigma}_{\beta_0}(z_0)_a - \tilde{\sigma}_\beta(h)_a)^2 \right\} , \qquad (63)$$

where we recall that $\tilde{\sigma}_\beta(A)_{ab} = \sigma_\beta(\sqrt{1 + \delta_{ab}} A_{ab})$ for any matrix $A \in \mathbb{R}^{T \times T}$. For the softmax activation, it is easy to see that the minimum is achieved at

$$p_{ab} = z_{0,ab} + \frac{\bar{a}}{\sqrt{1 + \delta_{ab}}} \qquad (64)$$

for any scalar $\bar{a}$. Thus, we have a continuum of global minima, and we need to pick the one closest in L2 distance to $z$, giving

$$p_{ab} = z_{0,ab} - \frac{2}{T^2}\sum_{c \leq d} \frac{z_{0,cd} - z_{cd}}{\sqrt{(1 + \delta_{cd})(1 + \delta_{ab})}} \qquad (65)$$

which provides striking simplifications in the equations, that reduce to

$$
\begin{cases}
q & = \frac{\hat{m}^2}{\hat{\Sigma}^2} J\left(\frac{\sqrt{\hat{q}}}{\hat{m}}, \frac{2}{\hat{m}}\right) \\
m & = \frac{\hat{m} J\left(\frac{\sqrt{\hat{q}}}{\hat{m}}, \frac{2}{\hat{m}}\right) - \tilde{\lambda} \partial_2 J\left(\frac{\sqrt{\hat{q}}}{\hat{m}}, \frac{2}{\hat{m}}\right) - \frac{\sqrt{\hat{q}}}{2} \partial_1 J\left(\frac{\sqrt{\hat{q}}}{\hat{m}}, \frac{2}{\hat{m}}\right)}{\hat{\Sigma}} \\
\Sigma \hat{\Sigma} & = \frac{\hat{m}}{2\sqrt{\hat{q}}} \partial_1 J\left(\frac{\sqrt{\hat{q}}}{\hat{m}}, \frac{2}{\hat{m}}\right) \\
\Sigma \hat{\Sigma} & = 2\alpha(L_{\text{in}} - 1) \\
\hat{m} \Sigma & = 2\alpha(L_{\text{in}} - 1) \\
\hat{q} \Sigma^2 & = 2\alpha(L_{\text{in}} - 1)(Q_0 - 2m + q)
\end{cases}
\tag{66}
$$

It is easy to see that instead, in the linear case, $p = z_0$, leading to the set of equations

$$
\begin{cases}
q & = \frac{\hat{m}^2}{\hat{\Sigma}^2} J\left(\frac{\sqrt{\hat{q}}}{\hat{m}}, \frac{2}{\hat{m}}\right) \\
m & = \frac{\hat{m} J\left(\frac{\sqrt{\hat{q}}}{\hat{m}}, \frac{2}{\hat{m}}\right) - \tilde{\lambda} \partial_2 J\left(\frac{\sqrt{\hat{q}}}{\hat{m}}, \frac{2}{\hat{m}}\right) - \frac{\sqrt{\hat{q}}}{2} \partial_1 J\left(\frac{\sqrt{\hat{q}}}{\hat{m}}, \frac{2}{\hat{m}}\right)}{\hat{\Sigma}} \\
\Sigma \hat{\Sigma} & = \frac{\hat{m}}{2\sqrt{\hat{q}}} \partial_1 J\left(\frac{\sqrt{\hat{q}}}{\hat{m}}, \frac{2}{\hat{m}}\right) \\
\Sigma \hat{\Sigma} & = 2\alpha L_{\text{in}} \\
\hat{m} \Sigma & = 2\alpha L_{\text{in}} \\
\hat{q} \Sigma^2 & = 2\alpha L_{\text{in}}(Q_0 - 2m + q).
\end{cases}
\tag{67}
$$

We thus see that, calling $\alpha_{\text{soft}} = (L_{\text{in}} - 1)\alpha$ and $\alpha_{\text{linear}} = L_{\text{in}}\alpha$, one get the same equations as the single-token case Erba et al. (2025).

This implies immediately Corollary B.2, giving in particular the expression for the interpolation threshold up to which the mapping discussed in this Appendix holds. It also implies immediately that the low-rank limit $\kappa_0 \to 0$ reduces to Result 2 of Erba et al. (2025) (modulo a rescaling).

# G    NON-FACTORIZED ATTENTION

The case of non-factorized attention falls in the formalism of Appendix E.1, with state evolution equations given by

$$
q = \frac{\hat{m}^2}{(\hat{\Sigma} + 4\tau)^2} \left(Q_0 + \frac{\hat{q}}{\hat{m}^2}\right)
$$

$$
m = \frac{Q_0 \hat{m}}{\hat{\Sigma} + 4\tau}
$$

$$
\Sigma = \frac{1}{\hat{\Sigma} + 4\tau}
$$

$$
\Sigma \hat{\Sigma} = 2\alpha L - 2\alpha \mathbb{E}_{z_0, z} \sum_{a=1}^{L_{\text{in}}} \frac{Q_0 z_a p_a - m z_{0,a} p_a}{Q_0 q - m^2}
\tag{68}
$$

$$
\Sigma \hat{m} = 2\alpha \mathbb{E}_{z_0, z} \sum_{a=1}^{L_{\text{in}}} \frac{q z_{0,a} p_a - m z_a p_a}{Q_0 q - m^2}
$$

$$
\hat{q} \Sigma^2 = 2\alpha \mathbb{E}_{z_0, z} \sum_{a=1}^{L_{\text{in}}} (p_a - z_a)^2
$$

with $\tau$ being the Frobenius regularization. The equations can be derived by noticing that

$$
\begin{aligned}
\Psi(\hat{\Sigma}, \hat{q}, \hat{m}) &= \frac{2}{d} \mathbb{E}_D \min_{T \in \mathcal{C}} \left[ \frac{1}{d^2} R(T) + \frac{\hat{\Sigma}}{4} \sum_{i=1}^d T_i^2 - \frac{1}{2} \sum_{i=1}^d T_i D_i \right] \\
&= \frac{1}{d} \mathbb{E}_D \min_{T \in \mathcal{C}} \left[ \frac{\hat{\Sigma} + 4\tau}{2} \sum_{i=1}^d T_i^2 - \sum_{i=1}^d T_i D_i \right] \\
&= -\frac{1}{2(\hat{\Sigma} + 4\tau)} \frac{1}{d} \mathbb{E}_D \sum_{i=1}^d T_i^2 \\
&= -\frac{Q_0 \hat{m}^2 + \hat{q}}{2(\hat{\Sigma} + 4\tau)}
\end{aligned}
\tag{69}
$$

while the loss-related equations are the same as in the case of the main text.

In the limit $\tau \to 0^+$, the last three equations reduce as in the case of Appendix F, giving a set of equations that can be solved and gives (in the noiseless case)

$$
||\hat{S} - S_0||_F^2 = Q_0 - 2m + q = Q_0(1 - 2\alpha) \tag{70}
$$

giving the strong recovery at $\alpha = 1/2$ and proving the claim that if $\kappa_0 \to 0^+$ learning for the non-factorized model happens only at scale $\alpha = \mathcal{O}(1)$, i.e. $n = \mathcal{O}(d^2)$.

# H    DERIVATION OF THE ERROR DECOMPOSITION OF SECTION 4

To begin with, the generalization error is given by

$$
\mathcal{I}(m, q) = \mathbb{E}_{z_0, z, \xi} ||\sigma_1(z_0 + \sqrt{\Delta/2}\zeta) - \sigma_2(z)||^2, \tag{71}
$$

where $(z_0, z) \sim \mathcal{N}\left(0, \begin{pmatrix} Q_0 & m \\ m & q \end{pmatrix}\right)$, $\zeta \sim \mathcal{N}(0, 1)$ and for simplicity we denote $\sigma_1 := \tilde{\sigma}_{\beta_0}$ to be the teacher activation and $\sigma_2 := \tilde{\sigma}_\beta$ to be the student activation.

We further assume that Claim B.1 is valid for any $n, d$ large enough, as specified by the following conjecture.

**Conjecture H.1.** *Let $\lambda > 0$, $\Delta \geq 0$ and consider $n, d \gg 1$ sufficiently large. Then with high probability the generalization error at the global minimum is given by*

$$
e_{\text{test}}(\hat{W}) = \mathcal{I}(m_0, q_0) + (\mathcal{I}(m^\star, q^\star) - \mathcal{I}(m_0, q_0))(1 + o_{n,d}(1)), \tag{72}
$$

*where $m_0, q_0$ independent of $n, d$ are the zeroth orders of $m^\star, q^\star$ (i.e. $m^\star = m_0(1 + o_{n,d}(1))$, $q^\star = q_0(1 + o_{n,d}(1))$) and $m^\star, q^\star$ are obtained as the global minimizer of*

$$
\Phi(\Sigma, m, q, \hat{\Sigma}, \hat{m}, \hat{q}) = \frac{\hat{q}\Sigma + 2\hat{m}m - \hat{\Sigma}q}{4} + \frac{n}{d^2} \mathcal{M}(\Sigma, m, q) - \frac{\hat{m}^2}{4\hat{\Sigma}} J\left(\frac{\sqrt{\hat{q}}}{\hat{m}}, \frac{2\tilde{\lambda}}{\hat{m}}\right). \tag{73}
$$

In another word, we conjecture that the generalization error is given by

$$
e_{\text{test}}(\hat{W}) \approx I(m^\star, q^\star), \tag{74}
$$

up to the first order and $m^\star, q^\star$ are obtained by minimizing (9) at large but finite $n, d$.

## H.1    SHARP MINIMUM: $\Sigma \ll 1$

When $\Sigma \ll 1$ we approximately have

$$
\mathcal{M}(\Sigma, m, q) \approx \mathbb{E}_{z_0, z}(||\sigma_1(z_0) - \sigma_2(z)||^2 - 4\Sigma||\mathcal{J}_{\sigma_2}(z)(z - z_0)||^2), \tag{75}
$$

where $z, z_0$ are regarded as $\frac{1}{2}T(T+1)-$dimensional vector and thus $\mathcal{J}_{\sigma_2} \in \mathbb{R}^{\frac{1}{2}T(T+1) \times \frac{1}{2}T(T+1)}$ denotes the derivative of $\sigma_1$.

Under the assumption that the cosine similarity between the teacher and the student is close to 1, i.e.,

$$q - \frac{m^2}{Q_0} \ll Q_0, \tag{76}$$

we can write $z_0 = \sqrt{\frac{\Delta}{2}}\zeta_1 + \sqrt{Q_0}\zeta_0$ and $z = \frac{m}{\sqrt{Q_0}}\zeta_0 + \sqrt{q - \frac{m^2}{Q_0}}\zeta$ with $\zeta, \zeta_0, \zeta_1 \sim \mathcal{N}(0,1)$ independent of each other and then expand (75) as

$$\mathbb{E}_{z_0, z} ||\sigma_1(z_0) - \sigma_2(z)|^2$$

$$\approx \mathbb{E}_{\zeta, \zeta_0, \zeta_1} \left\| \sigma_1 \left( \sqrt{\frac{\Delta}{2}}\zeta_1 + \sqrt{Q_0}\zeta_0 \right) - \sigma_2 \left( \frac{m}{\sqrt{Q_0}}\zeta_0 \right) + \mathcal{J}_{\sigma_2} \left( \frac{m}{\sqrt{Q_0}}\zeta_0 \right) \sqrt{q - \frac{m^2}{Q_0}}\zeta \right\|^2$$

$$= \mathbb{E}_{\zeta_0, \zeta_1} \left\| \sigma_1 \left( \sqrt{\frac{\Delta}{2}}\zeta_1 + \sqrt{Q_0}\zeta_0 \right) - \sigma_2(\frac{m}{\sqrt{Q_0}}\zeta_0) \right\|^2 + \left( q - \frac{m^2}{Q_0} \right) \mathbb{E}_{\zeta_0} \left\| \mathcal{J}_{\sigma_2} \left( \frac{m}{\sqrt{Q_0}}\zeta_0 \right) \right\|_F^2 \tag{77}$$

and

$$\mathbb{E}_{z_0, z} ||\mathcal{J}_{\sigma_2}(z)(z - z_0)||^2$$

$$= \mathbb{E}_{\zeta, \zeta_0, \zeta_1} \left\| \mathcal{J}_{\sigma_2} \left( \sqrt{\frac{\Delta}{2}}\zeta_1 + \sqrt{Q_0}\zeta_0 \right) \left( (\sqrt{Q_0} - \frac{m}{\sqrt{Q_0}})\zeta_0 + \sqrt{\frac{\Delta}{2}}\zeta_1 - \sqrt{q - \frac{m^2}{Q_0}}\zeta \right) \right\|^2$$

$$\approx \mathbb{E}_{\zeta_0} ||\mathcal{J}_{\sigma_2}(\sqrt{Q_0 + \Delta/2}\zeta_0)||_F^2 \left( q - \frac{m^2}{Q_0} \right) + \mathbb{E}_{\zeta_0, \zeta_1} \left\| \mathcal{J}_{\sigma_2} \left( \sqrt{\frac{\Delta}{2}}\zeta_1 + \sqrt{Q_0}\zeta_0 \right) \left( \left( \sqrt{Q_0} - \frac{m}{\sqrt{Q_0}} \right) \zeta_0 + \sqrt{\frac{\Delta}{2}}\zeta_1 \right) \right\|^2, \tag{78}$$

which gives

$$\mathcal{M}(\Sigma, m, q) \approx \kappa_0(m) + \kappa_1(m)(q - \frac{m^2}{Q_0}) - 4\Sigma\kappa_2(m) \tag{79}$$

up to the first order of $\Sigma$ and $q - \frac{m^2}{Q_0}$, where

$$\kappa_0(m) := \mathbb{E}_{\zeta_0, \zeta_1} \left\| \sigma_1 \left( \sqrt{Q_0}\zeta_0 + \sqrt{\frac{\Delta}{2}}\zeta_1 \right) - \sigma_2 \left( \frac{m}{\sqrt{Q_0}}\zeta_1 \right) \right\|^2 = \mathcal{I}(m, m^2/Q_0), \tag{80}$$

$$\kappa_1(m) := \mathbb{E}_\zeta \left\| \mathcal{J}_{\sigma_2} \left( \frac{m}{\sqrt{Q_0}}\zeta \right) \right\|_F^2 \tag{81}$$

and

$$\kappa_2(m) := \mathbb{E}_{\zeta, \zeta_0} \left\| \mathcal{J}_{\sigma_2} \left( \sqrt{Q_0}\zeta_0 + \sqrt{\frac{\Delta}{2}}\zeta \right) \left( \left( \sqrt{Q_0} - \frac{m}{\sqrt{Q_0}} \right) \zeta_0 + \sqrt{\frac{\Delta}{2}}\zeta \right) \right\|_F^2. \tag{82}$$

Then the saddle point equations for (9) are

$$\begin{cases} \hat{\Sigma} = 4\alpha\kappa_1 \\ \hat{m} = 2\alpha(\kappa_1 \frac{2m}{Q_0} + \kappa_0'(m)) \\ \hat{q} = 4\alpha\kappa_2 \\ m = \partial_{\hat{m}} \frac{\hat{m}^2}{2\hat{\Sigma}} J(\frac{\sqrt{\hat{q}}}{\hat{m}}, \frac{2\tilde{\lambda}}{\hat{m}}) \\ q = \left( \frac{\hat{m}}{\hat{\Sigma}} \right)^2 J(\frac{\sqrt{\hat{q}}}{\hat{m}}, \frac{2\tilde{\lambda}}{\hat{m}}) \\ \Sigma = \frac{\hat{m}^2}{\hat{\Sigma}} \partial_{\hat{q}} J(\frac{\sqrt{\hat{q}}}{\hat{m}}, \frac{2\tilde{\lambda}}{\hat{m}}). \end{cases} \tag{83}$$

At the leading order we have $J \approx Q_0$ because the estimator is given by $\hat{W}^T\hat{W}/\sqrt{pd} = \text{ReLU}(\frac{\hat{m}}{\hat{\Sigma}}S_0 + \frac{\sqrt{\hat{q}}}{\hat{\Sigma}}Z - \frac{2\lambda}{\hat{\Sigma}})$ in distribution and $J$ describes the second moment of $\text{ReLU}(S_0 + \frac{\sqrt{\hat{q}}}{\hat{m}}Z - \frac{2\lambda}{\hat{m}})$. Then $m^2 \approx qQ_0$ is equivalent to $J \approx Q_0$. One can also check at if we use $J \approx Q_0$, we have $\text{Tr}[\hat{W}^T\hat{W}S_0/\sqrt{pd}] = \frac{\hat{m}}{\hat{\Sigma}}Q_0 = m$ in the leading order.

Then taking the first two equations of (83) into the last three equations of (83) gives the first order solutions

$$
\begin{cases}
m_0 & = \frac{2\kappa_1 m_0/Q_0 + \kappa_0'(m_0)}{2\kappa_1} Q_0, \\
q_0 & = \left( \frac{2\kappa_1 m_0/Q_0 + \kappa_0'(m_0)}{2\kappa_1} \right)^2 Q_0, \\
\Sigma & = 0
\end{cases}
\tag{84}
$$

consistent with the ansatzs $q_0 = \frac{m_0^2}{Q_0}$ and $\Sigma \ll 1$. Then the leading order solution is given by $q_0 = \frac{m_0^2}{Q_0}$ with $m_0$ satisfying

$$
\kappa_0'(m_0) = 0,
\tag{85}
$$

or equivalently

$$
m_0 = \arg\inf_m \mathcal{I}(m, m^2/Q_0).
\tag{86}
$$

This gives the zeroth order generalization error

$$
e_0 = \inf_m \mathcal{I}(m, m^2/Q_0).
\tag{87}
$$

Now we consider the first order correction. The excess risk is given by

$$
\begin{aligned}
e_1 &:= \frac{\partial \mathcal{I}(m,q)}{\partial m}\Big|_{m_0,q_0} m_1 + \frac{\partial \mathcal{I}(m,q)}{\partial q}\Big|_{m_0,q_0} q_1 \\
&= \frac{\chi}{\eta}(q_1 - 2m_1),
\end{aligned}
\tag{88}
$$

where $m_1, q_1$ are the first order corrections, $\eta := \frac{\hat{m}}{\hat{\Sigma}} = \frac{m_0}{Q_0}$ and

$$
\chi := \mathbb{E}\left[ \mathrm{Tr}\left( \mathcal{J}_{\sigma_1}\left( z_0 + \sqrt{\frac{\Delta}{2}}\xi \right) \mathcal{J}_{\sigma_2} (\eta z_0)^T \right) \right].
\tag{89}
$$

The first order corrections of fourth and fifth equations of (83) are

$$
\begin{aligned}
m_1 &= \frac{\hat{m}}{\hat{\Sigma}}(J(\delta,\epsilon) - Q_0) - \frac{1}{2\hat{\Sigma}}(\sqrt{\hat{q}}\partial_1 J(\delta,\epsilon) + \tilde{\lambda}\partial_2 J(\delta,\epsilon)) \\
&= \eta(J(\delta,\epsilon) - Q_0) - \frac{\eta}{2}(\delta\partial_1 J(\delta,\epsilon) + \epsilon\partial_2 J(\delta,\epsilon))
\end{aligned}
\tag{90}
$$

and

$$
q_1 = \left( \frac{\hat{m}}{\hat{\Sigma}} \right)^2 (J(\delta,\epsilon) - Q_0) = \eta^2(J(\delta,\epsilon) - Q_0),
\tag{91}
$$

and thus

$$
e_1 = \eta\chi((\delta\partial_1 + \epsilon\partial_2 - 1)J(\delta,\epsilon) + Q_0),
\tag{92}
$$

where we define

$$
\delta := \frac{\sqrt{\hat{q}}}{\hat{m}} = \frac{\sqrt{\kappa_2(m_0)}}{\sqrt{\alpha}\eta\kappa_1(m_0)}
\tag{93}
$$

and

$$
\epsilon := \frac{2\tilde{\lambda}}{\hat{m}} = \frac{\tilde{\lambda}}{2\alpha\kappa_1(m_0)\eta}.
\tag{94}
$$

Therefore, the excess risk can be written as

$$
e_1 := \eta\chi((\delta\partial_1 + \epsilon\partial_2 - 1)J(\delta,\epsilon) + Q_0).
\tag{95}
$$

Finally, assume that the spectrum is given by the BBP approximation, i.e., for the under-regularization case

$$
J(\delta,\epsilon) = \frac{1}{d}\sum_{i=1}^{K(\delta)} \left( s_i + \frac{\delta^2}{s_i} - \epsilon \right)^2 + \delta^2 \int_{\epsilon/\delta}^{2} \mu_{\mathrm{sc}}(x)(x - \epsilon/\delta)^2
\tag{96}
$$

with a cutoff $K(\delta) \ll d$ satisfying $s_{K(\delta)} = \delta$, or for the over-regularization case

$$J(\delta, \epsilon) = \frac{1}{d} \sum_{i=1}^{K(\delta, \epsilon)} (s_i + \frac{\delta^2}{s_i} - \epsilon)^2 \tag{97}$$

with a cutoff $K(\delta, \epsilon) \ll d$ satisfying $s_{K(\delta, \epsilon)} + \frac{\delta^2}{s_{K(\delta, \epsilon)}} - \epsilon = 0$. Then we can borrow Result 3 of Defilippis et al. (2025) to write

$$e_1 = \eta \chi \left( \underbrace{\delta^2 \int_{\epsilon/\delta}^2 \mu_{sc}(\mathrm{d}x) \left( x - \frac{\epsilon}{\delta} \right)^2 + \frac{1}{d} \delta K'(\delta)(2\delta - \epsilon)^2}_{\substack{\text{overfitting} \\ \text{(learned noise)}}} + \underbrace{\frac{1}{d} \sum_{i=K(\delta)+1}^{d} s_i^2}_{\substack{\text{underfitting} \\ \text{(not learned features)}}} + \underbrace{\frac{1}{d} \sum_{i=1}^{K(\delta)} \left[ \left( \frac{\delta^2}{s_i} - \epsilon \right)^2 + \frac{\delta^2}{s_i} \left( s_i + \frac{\delta^2}{s_i} - \epsilon \right) \right]}_{\substack{\text{approximation error} \\ \text{for learned features}}} \right) \tag{98}$$

for the under-regularization case and

$$e_1 = \eta \chi \left( \underbrace{\frac{1}{d} \sum_{i=K(\delta, \epsilon)+1}^{d} s_i^2}_{\substack{\text{underfitting} \\ \text{(not learned features)}}} + \underbrace{\frac{1}{d} \sum_{i=1}^{K(\delta, \epsilon)} \left[ \left( \frac{\delta^2}{s_i} - \epsilon \right)^2 + \frac{\delta^2}{s_i} \left( s_i + \frac{\delta^2}{s_i} - \epsilon \right) \right]}_{\substack{\text{approximation error} \\ \text{for learned features}}} \right) \tag{99}$$

for the over-regularization case. This finishes the derivation of the error decomposition for the case $\Sigma \ll 1$.

**When is it the global minimum?** One thing to notice is that the plateau (87) is not necessarily equal to the oracle generalization error

$$\min_{m,q:m^2 \leq qQ_0} \mathcal{I}(m, q). \tag{100}$$

Thus it only describes one saddle point rather than the global minimum.

Here we only point out that a sufficient condition for (87) to be the oracle generalization error is when the activations are conservative monotone operators, i.e., if

$$\sigma_1 = \nabla \Phi_1, \ \sigma_2 = \nabla \Phi_2 \tag{101}$$

for functions $\Phi_1, \Phi_2$ which are both convex or both concave. This can be easily proven by

$$\begin{aligned} \frac{\partial \mathcal{I}(m, q)}{\partial m} &= -\mathbb{E}_{z_0, z} \frac{\partial \sigma_1(z_0 + \sqrt{\Delta/2}\xi)^T \sigma_2(z)}{\partial m} \\ &= -\sum_{k=1}^{d} \mathbb{E} \left[ \frac{\partial^2}{\partial y_k \partial z_k} \left( \sigma_1(y)^T \sigma_2(z) \right) \right] \\ &= -\mathbb{E} \left[ \mathrm{Tr} \left( \mathcal{J}_{\sigma_1}(y) \left( \mathcal{J}_{\sigma_1}(z) \right)^T \right) \right] \leq 0, \end{aligned} \tag{102}$$

where we denote $y := z_0 + \sqrt{\Delta/2}\zeta$ and use the Stein's lemma. The last inequality is because if $\sigma_1, \sigma_2$ are conservative monotone, $\mathcal{J}_{\sigma_1}, \mathcal{J}_{\sigma_1}$ are positive/negative semi-definite. Then the oracle generalization error is achieved when $m = \sqrt{qQ_0}$, which proves that $e_0$ is equal to the oracle generalization error.

The condition can be relaxed to row-wise or column-wise conservative monotone, because we can write

$$\mathcal{I}(m, q) = \sum_{\ell=1}^{d} \mathbb{E}_{z_0, z, \zeta}(\sigma_{1\ell}(z_0 + \sqrt{\Delta/2}\zeta) - \sigma_{2\ell}(z))^2. \tag{103}$$

Then the softmax activation satisfies this condition because it is row-wise conservative monotone: $\mathrm{softmax}(x) = \nabla \log \sum_i e^{x_i}$.

**The potential global minimum** In general we can write the SE at $\Sigma \ll 1$ as

$$\begin{cases} \hat{\Sigma} = 4\alpha \frac{\partial \mathcal{I}(m,q)}{\partial m} \\ \hat{m} = -2\alpha \frac{\partial \mathcal{I}(m,q)}{\partial q} \\ \hat{q} = 4\alpha\kappa_2(m,q) \\ m = \partial_{\hat{m}} \frac{\hat{m}^2}{2\hat{\Sigma}} J(\frac{\sqrt{\hat{q}}}{\hat{m}}, \frac{2\lambda}{\hat{m}}) \\ q = (\frac{\hat{m}}{\hat{\Sigma}})^2 J(\frac{\sqrt{\hat{q}}}{\hat{m}}, \frac{2\lambda}{\hat{m}}) \\ \Sigma = \frac{\hat{m}^2}{\hat{\Sigma}} \partial_{\hat{q}} J(\frac{\sqrt{\hat{q}}}{\hat{m}}, \frac{2\lambda}{\hat{m}}), \end{cases} \tag{104}$$

where

$$\kappa_2(m,q) := \mathbb{E}_{z,z_0} ||\mathcal{J}_{\sigma_2}(z)(z - z_0)||^2. \tag{105}$$

For the global minimum, the plateau is given by

$$e_0 = \inf_{m,q:m^2 \leq qQ_0} \mathcal{I}(m,q). \tag{106}$$

Then if the ansatz $m^2 = qQ_0$ is not valid at the global minimum, we must have

$$\frac{\partial \mathcal{I}(m,q)}{\partial m}\Big|_{m_0,q_0} = \frac{\partial \mathcal{I}(m,q)}{\partial q}\Big|_{m_0,q_0} = 0, \tag{107}$$

which gives the first order solution $m_0, q_0$. Then we have $\hat{q}_0 = 4\alpha\kappa_2(m_0, q_0)$ and $\hat{m}_0, \hat{\Sigma}_0$ are solved self-consistently from

$$m_0 = \partial_{\hat{m}_0} \frac{\hat{m}_0^2}{2\hat{\Sigma}_0} J\left(\frac{\sqrt{\hat{q}_0}}{\hat{m}_0}, \frac{2\lambda}{\hat{m}_0}\right), \quad q_0 = \left(\frac{\hat{m}_0}{\hat{\Sigma}_0}\right)^2 J\left(\frac{\sqrt{\hat{q}_0}}{\hat{m}_0}, \frac{2\lambda}{\hat{m}_0}\right). \tag{108}$$

The excess risk is then given by

$$e_1 := \frac{1}{2} \frac{\partial^2 \mathcal{I}(m,q)}{\partial m^2}\Big|_{m_0,q_0} m_1^2 + \frac{1}{2} \frac{\partial^2 \mathcal{I}(m,q)}{\partial q^2}\Big|_{m_0,q_0} q_1^2 + \frac{\partial^2 \mathcal{I}(m,q)}{\partial m \partial q}\Big|_{m_0,q_0} m_1 q_1, \tag{109}$$

where $m_1, q_1$ are the first order corrections. It is left as the future work to explicitly calculate the excess risk in this scenario.

## H.2 FLAT MINIMUM: $\Sigma \gg 1$

When $\Sigma \gg 1$, we approximately have

$$\mathcal{M}(\Sigma, m, q) = \frac{1}{2\Sigma} \mathcal{K}(m,q), \tag{110}$$

where

$$\mathcal{K}(m,q) := \mathbb{E}_{z_0,z,\zeta} \inf_{h \in \arg\inf ||\sigma_1(z_0 + \sqrt{\Delta/2}\zeta) - \sigma_2(\cdot)||^2} ||h - z||^2. \tag{111}$$

For example, when $\sigma_2(x) = \sigma_1(\beta^{-1}x)$ and $\sigma_1$ is a one-to-one mapping, we have

$$\mathcal{M}(\Sigma, m, q) = \mathbb{E}_{z_0,z} \frac{1}{2\Sigma} \sum_{a \leq b}^{T} (\beta z_{0,ab} - z_{ab})^2 = \frac{T(T+1)}{4\Sigma} (\beta^2 Q_0 + \beta^2 \frac{\Delta}{2} + q - 2\beta m). \tag{112}$$

When $\sigma_2(x) = \sigma_1(\beta^{-1}x)$ and $\sigma_1(z) = \text{softmax}(\{\sqrt{1 + \delta_{ab}} z_{ab}\}_{ab})$, there is a redundant degree of freedom, so we have

$$\mathcal{M}(\Sigma, m, q) = \mathbb{E}_{z_0,z} \inf_{\xi} \frac{1}{2\Sigma} \sum_{a \leq b}^{T} (\beta z_{0,ab} + \frac{\xi}{\sqrt{1 + \delta_{ab}}} - z_{ab})^2$$

$$= \frac{1}{2\Sigma} \left[\frac{1}{2} T(T+1) - 1\right] (\beta^2 Q_0 + \beta^2 \frac{\Delta}{2} + q - 2\beta m). \tag{113}$$

By using (75) the SE reduces to

$$
\begin{cases}
\hat{\Sigma} = \frac{2\alpha}{\Sigma} \frac{\partial \mathcal{K}(m,q)}{\partial q} \\
\hat{m} = -\frac{\alpha}{\Sigma} \frac{\partial \mathcal{K}(m,q)}{\partial m} \\
\hat{q} = 2\alpha \frac{\mathcal{K}(m,q)}{\Sigma^2} \\
m = \partial_{\hat{m}} \frac{\hat{m}^2}{2\hat{\Sigma}} J(\frac{\sqrt{\hat{q}}}{\hat{m}}, \frac{2\lambda}{\hat{m}}) \\
q = (\frac{\hat{m}}{\hat{\Sigma}})^2 J(\frac{\sqrt{\hat{q}}}{\hat{m}}, \frac{2\lambda}{\hat{m}}) \\
\Sigma = \frac{\hat{m}^2}{\hat{\Sigma}} \partial_{\hat{q}} J(\frac{\sqrt{\hat{q}}}{\hat{m}}, \frac{2\lambda}{\hat{m}}).
\end{cases}
\tag{114}
$$

Under the ansatz $J \approx Q_0$, we have

$$
m_0 = \eta Q_0, \ q_0 = \eta^2 Q_0
\tag{115}
$$

at the leading order by the fourth and fifth equations of (114), satisfying $m_0^2 = q_0 Q_0$, where we denote $\eta := \frac{\hat{m}}{\hat{\Sigma}} = \frac{m_0}{Q_0}$. They are solved self-consistently with

$$
\eta = -\frac{1}{2} \left[ \frac{\partial \mathcal{K}(m,q)}{\partial q} \Big|_{m_0,q_0} \right]^{-1} \frac{\partial \mathcal{K}(m,q)}{\partial m} \Big|_{m_0,q_0},
\tag{116}
$$

which is obtained by combining the first and the second equations of (114). This gives the plateau $e_0 = \mathcal{I}(m_0, q_0)$. Note that in this case, the plateau $\mathcal{I}(m_0, q_0)$ is non-optimal but does not depend on $n, d, \lambda$.

Then we can similarly define the excess risk as

$$
\begin{aligned}
e_1 :&= \frac{\partial \mathcal{I}(m,q)}{\partial m} \Big|_{m_0,q_0} m_1 + \frac{\partial \mathcal{I}(m,q)}{\partial q} \Big|_{m_0,q_0} q_1 \\
&= -2\chi m_1 + \chi' q_1
\end{aligned}
\tag{117}
$$

where $m_1, q_1$ are the first order correction. $\chi$ is defined the same way as (89) and

$$
\chi' := \mathbb{E} \left[ \| \mathcal{J}_{\sigma_2}(\eta z_0) \|_F^2 + \sum_{i=1}^{d^2} (\sigma_{2i}(\eta z_0) - \sigma_{1i}(z_0)) \Delta \sigma_i(\eta z_0) \right].
\tag{118}
$$

The first order corrections are the same as the previous case

$$
m_1 = \eta(J(\delta,\epsilon) - Q_0) - \frac{\eta}{2}(\delta \partial_1 J(\delta,\epsilon) + \epsilon \partial_2 J(\delta,\epsilon))
\tag{119}
$$

and

$$
q_1 = \eta^2(J(\delta,\epsilon) - Q_0),
\tag{120}
$$

which gives

$$
\begin{aligned}
e_1 &= (\eta \chi' - \chi)\eta(J(\delta,\epsilon) - Q_0) + \eta\chi((\delta \partial_1 + \epsilon \partial_2 - 1)J(\delta,\epsilon) + Q_0) \\
&= \eta\tilde{\chi}(J(\delta,\epsilon) - Q_0) + \eta\chi((\delta \partial_1 + \epsilon \partial_2 - 1)J(\delta,\epsilon) + Q_0),
\end{aligned}
\tag{121}
$$

where

$$
\tilde{\chi} := \frac{1}{2}\sqrt{1 + 2\eta^2} \frac{\partial \mathcal{I}(m,q)}{\partial \ell} \Big|_{m_0,q_0}
\tag{122}
$$

with $\ell$ the direction along $m^2 = qQ_0$. There is an additional term compared with the original error decomposition, with coefficient proportional to the derivative along $m^2 = qQ$ because

$$
\begin{aligned}
\frac{\partial \mathcal{I}(m,q)}{\partial \ell} &= \frac{1}{\sqrt{Q_0^2 + 4m_0^2}} \left( Q_0 \frac{\partial \mathcal{I}(m,q)}{\partial m} \Big|_{m_0,q_0} + 2m_0 \frac{\partial \mathcal{I}(m,q)}{\partial q} \Big|_{m_0,q_0} \right) \\
&= \frac{2Q_0}{\sqrt{Q_0^2 + 4m_0^2}} (\eta\chi' - \chi).
\end{aligned}
\tag{123}
$$

This derivative becomes zero if the plateau is equal to (87), and thus evaluates how suboptimal the plateau is.

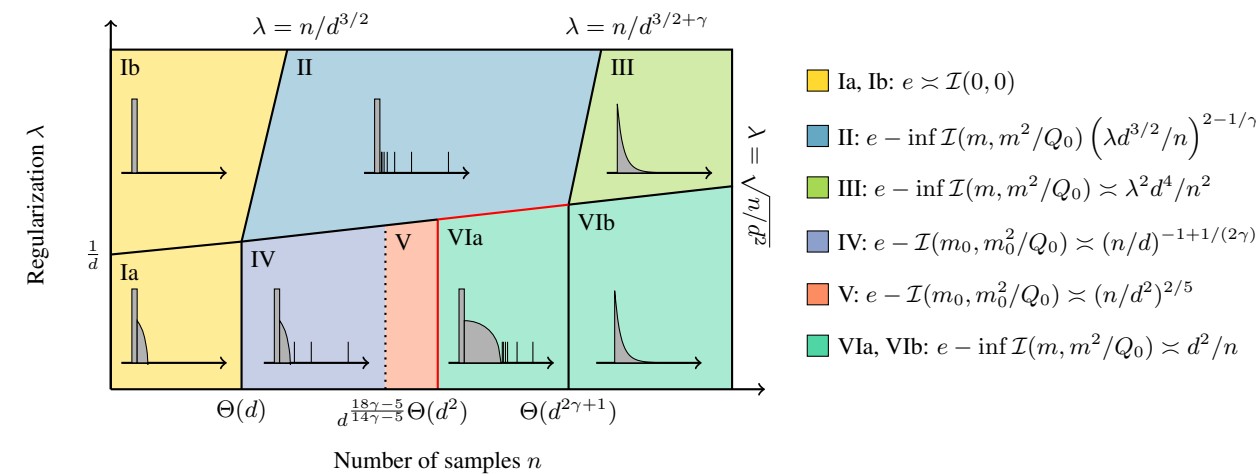

Figure 5: Excess risk rates of as a function of $n$ and $\lambda(n, d)$, with a sketch of the corresponding spectral properties of the learned weights.

Finally, to obtain $\delta$ and $\epsilon$, we can use the second and third equations of (114) to obtain

$$\alpha \delta^2 = \alpha \frac{\hat{q}}{\hat{m}^2} = \left[ \frac{\partial \mathcal{K}(m, q)}{\partial m}\big|_{m_0, q_0} \right]^{-2} \mathcal{K}(m_0, q_0). \tag{124}$$

Then using the first and the last equations of (114), we can obtain

$$2\alpha \frac{\partial \mathcal{K}(m, q)}{\partial q}\big|_{m_0, q_0} = \Sigma \hat{\Sigma} = \frac{1}{\delta} \partial_1 J(\delta, \epsilon). \tag{125}$$

$\delta, \epsilon$ are solved from (124) and (125).

Similarly, under the BBP approximation, we can write the error decomposition as

$$e_1 = \eta\chi \left( \underbrace{\delta^2 \int_{\epsilon/\delta}^2 \mu_{\mathrm{sc}}(\mathrm{d}x)\left(x - \frac{\epsilon}{\delta}\right)^2 + \frac{1}{d}\delta K'(\delta)(2\delta - \epsilon)^2}_{\substack{\text{overfitting} \\ \text{(learned noise)}}} + \underbrace{\frac{1}{d}\sum_{i=K(\delta)+1}^d s_i^2}_{\substack{\text{underfitting} \\ \text{(not learned features)}}} + \underbrace{\frac{1}{d}\sum_{i=1}^{K(\delta)} \left[ \left(\frac{\delta^2}{s_i} - \epsilon\right)^2 + \frac{\delta^2}{s_i}\left(s_i + \frac{\delta^2}{s_i} - \epsilon\right) \right]}_{\substack{\text{approximation error} \\ \text{for learned features}}} \right)$$

$$+ \eta\tilde{\chi} \left( \underbrace{\frac{1}{d}\sum_{i=1}^{K(\delta)}\left(s_i + \frac{\delta^2}{s_i} - \epsilon\right)^2 + \delta^2 \int_{\epsilon/\delta}^2 \mu_{\mathrm{sc}}(x)(x - \epsilon/\delta)^2 - \frac{1}{d}\sum_{i=1}^d s_i^2}_{\text{mismatch of the second moment}} \right) \tag{126}$$

for the under-regularization case and

$$e_1 = \eta\chi \left( \underbrace{\frac{1}{d}\sum_{i=K(\delta,\epsilon)+1}^d s_i^2}_{\substack{\text{underfitting} \\ \text{(not learned features)}}} + \underbrace{\frac{1}{d}\sum_{i=1}^{K(\delta,\epsilon)}\left[\left(\frac{\delta^2}{s_i} - \epsilon\right)^2 + \frac{\delta^2}{s_i}\left(s_i + \frac{\delta^2}{s_i} - \epsilon\right)\right]}_{\substack{\text{approximation error} \\ \text{for learned features}}} \right) + \eta\tilde{\chi}\left( \underbrace{\frac{1}{d}\sum_{i=1}^{K(\delta,\epsilon)}\left(s_i + \frac{\delta^2}{s_i} - \epsilon\right)^2 - \frac{1}{d}\sum_{i=1}^d s_i^2}_{\text{mismatch of the second moment}} \right) \tag{127}$$

for the over-regularization case. This finishes the derivation of the error decomposition for the case $\Sigma \ll 1$.

## I   ERROR SCALINGS FOR POWER-LAW TEACHERS

In this section we assume that the teacher has a power-law spectrum, i.e., $S_0$ has eigenvalues $\{\sqrt{d}i^{-\gamma}\}_{i=1}^p$ with $p \ll d$. The overall phase diagram is shown in Figure 5, which is the same as that in Defilippis et al. (2025). The detailed derivation is given in the following.

## I.1 SHARP MINIMUM: $\Sigma \ll 1$

**Phase Ib: Trivial phase** $\lambda \gg \max\left(\sqrt{\frac{n}{d^2}}, \frac{n}{d^{3/2}}\right)$   According to Defilippis et al. (2025), in this phase we have $J(\delta, \epsilon) \approx 0$. In this case (83) is still true, but gives the leading order solution

$$m, q, \Sigma = 0, \ \hat{\Sigma} = 4\alpha\kappa_1(0), \ \hat{m} = 2\alpha\kappa_0'(0), \ \hat{q} = 4\alpha\kappa_2(0). \tag{128}$$

This is consistent because $\frac{\lambda}{\hat{m}} \gg \sqrt{d}$. Thus we obtain the trivial generalization error $\mathcal{I}(0, 0)$.

**Phase II: Over-regularization phase** $\max\left(\sqrt{\frac{n}{d^2}}, \frac{n}{d^{\gamma+\frac{3}{2}}}\right) \ll \lambda \ll \frac{d^{3/2}}{n}$   According to Defilippis et al. (2025), in this phase we have

$$J(\delta, \epsilon) \approx Q_0 + \left(\frac{\gamma+1}{\gamma-1} - \frac{1}{2\gamma-1}\right)\left(\frac{\epsilon}{\sqrt{d}}\right)^{\frac{2\gamma-1}{\gamma}} - \frac{2\epsilon}{\sqrt{d}}\mathbf{1}_{\gamma>1}. \tag{129}$$

Taking it into (95), we have

$$e_1 = \frac{2\eta\chi\gamma}{2\gamma-1}\left(\lambda\frac{d^{3/2}}{2\kappa_1 n}\right)^{\frac{2\gamma-1}{\gamma}}. \tag{130}$$

The condition $\Sigma = \frac{\epsilon}{\delta}\partial_1 J(\delta, \epsilon) \ll 1$ is satisfied.

**Phase III: Intermediate over-regularization phase** $\sqrt{\frac{n}{d^2}} \ll \lambda \ll \frac{n}{d^{\gamma+3/2}}$   According to Defilippis et al. (2025), in this phase we have

$$J(\delta, \epsilon) \approx Q_0 - \epsilon p^{\min(\gamma,1)-1} + \lambda^2\epsilon^2. \tag{131}$$

Taking it into (95), we have

$$e_1 = \frac{\eta\chi\gamma\lambda^2 d^4}{4\kappa_1^2 n^2}. \tag{132}$$

The condition $\Sigma = \frac{\epsilon}{\delta}\partial_1 J(\delta, \epsilon) \ll 1$ is satisfied.

**Phase VI: Large-sample phase** $\lambda \ll \sqrt{\frac{n}{d^2}}$ **and** $n \gg d^{2\gamma+1}$   According to Defilippis et al. (2025), in this phase we have

$$J(\delta, \epsilon) = Q_0 + \frac{1}{2}\delta^2(1 + 4p/d) \approx Q_0 + \frac{\delta^2}{2}, \tag{133}$$

or

$$J(\delta, \epsilon) = Q_0 + \frac{1}{2}\delta^2 + C(\gamma)\left(\frac{\delta}{\sqrt{d}}\right)^{2-\frac{1}{\gamma}}. \tag{134}$$

Taking it into (95), we have

$$e_1 = \frac{\Delta d^2}{4\kappa_1 n}(1 + 4c) \quad \text{or} \quad \frac{\Delta d^2}{4\kappa_1 n}. \tag{135}$$

The condition $\Sigma = \frac{\epsilon}{\delta}\partial_1 J(\delta, \epsilon) = \Theta(1/\alpha) \ll 1$ is satisfied.

## I.2 FLAT MINIMUM: $\Sigma \gg 1$

**Phase Ia: Trivial phase** $n \ll d$ **and** $\lambda \ll \sqrt{\frac{n}{d^2}}$   According to Defilippis et al. (2025), in this phase we have

$$J(\delta, \epsilon) \approx \int_\epsilon^\delta \mu_{\rm sc}(x/\delta)/\delta(x-\epsilon)^2 dx \approx \delta^2\frac{16t^{7/2}}{105\pi}. \tag{136}$$

(124) and (125) are still true, and give $\delta = \sqrt{\frac{\Delta_1}{4\alpha}}$ and $\delta^2\frac{16t^{5/2}}{15\pi} = \Delta_2$, with constants defined as

$$\Delta_1 := 8\left[\frac{\partial\mathcal{K}(m,q)}{\partial m}\Big|_{m_0,q_0}\right]^{-2}\mathcal{K}(m_0,q_0), \ \Delta_2 := \Delta_1\frac{\partial\mathcal{K}(m,q)}{\partial q}\Big|_{m_0,q_0}. \tag{137}$$

Thus we have $J \approx 0$, which gives

$$m_0, q_0 = 0 \tag{138}$$

at the leading order. Then we obtain the trivial generalization error $\mathcal{I}(0, 0)$.

**Phase IV and V: Benign and harmful overfitting phase** $\lambda \ll \sqrt{\frac{n}{d^2}}$ **and** $d \ll n \ll d^2$ According to Defilippis et al. (2025), in this phase we have $J(\delta, \epsilon) \approx J_1(\delta, \epsilon) + J_2(\delta, \epsilon)$, where

$$J_2(\delta, \epsilon) := \int_\epsilon^\delta \mu_{\mathrm{sc}}(x/\delta)/\delta(x-\epsilon)^2 dx \approx \delta^2 \frac{16t^{7/2}}{105\pi} \tag{139}$$

with $t := 2 - \frac{\epsilon}{\delta} \ll 1$ and

$$J_1(\delta, \epsilon) : \approx Q_0 + \left(\frac{\delta}{\sqrt{d}}\right)^{2-\frac{1}{\gamma}} \left(-\frac{1}{2\gamma-1} + (\epsilon/\delta)^2 + 2 - 2\frac{\epsilon}{\delta}\frac{1}{1-\gamma} - \frac{2}{1+\gamma}\frac{\epsilon}{\delta} + \frac{1}{1+2\gamma}\right)$$
$$- \mathbf{1}_{\gamma>1}\zeta(\gamma)\frac{2\epsilon}{\sqrt{d}}. \tag{140}$$

Then we have

$$\delta\partial_1 J(\delta, \epsilon) \approx \delta^2 \frac{16t^{5/2}}{15\pi}. \tag{141}$$

Taking it into (124) and (125), we obtain $\delta = \sqrt{\frac{\Delta_1}{4\alpha}}$ and $\delta^2 \frac{16t^{5/2}}{15\pi} = \Delta_2$. Finally (121) gives the excess risk

$$e_1 = \frac{24\eta\chi\gamma^3}{4\gamma^3+4\gamma^2-\gamma-1}\left(\frac{d\Delta_1}{4n}\right)^{1-\frac{1}{2\gamma}} + \frac{\eta^2\chi'\Delta_2}{7}\left(\frac{15\pi\Delta_2}{4\Delta_1}\right)^{2/5}\left(\frac{n}{d^2}\right)^{2/5}. \tag{142}$$

The condition

$$\Sigma = \frac{\epsilon}{2\delta}\partial_1 J(\delta, \epsilon) \approx \frac{\Delta_2}{2\lambda\delta} \gg 1 \tag{143}$$

is satisfied because we have $\lambda \ll \sqrt{\alpha}$ in this phase.

**Interpolation peak** According to Defilippis et al. (2025), at the interpolation peak the spectrum is mainly determined by the noise, which gives

$$J(\delta, \epsilon) \approx Q_0 + \frac{1}{2}\delta^2 - \frac{8}{3\pi}\epsilon\delta \quad \text{or} \quad \frac{1}{2}\delta^2 - \frac{8}{3\pi}\epsilon\delta \tag{144}$$

at the leading order. From the fourth and fifth equations of (114) we have

$$m = \frac{\eta}{2}(2 - \delta\partial_1 - \epsilon\partial_2)J(\delta, \epsilon) = Q_0 \quad \text{or} \quad 0 \tag{145}$$

and

$$q = \eta^2 J(\delta, \epsilon), \tag{146}$$

where $\eta := \frac{\hat{m}}{\hat{\Sigma}}$ is determined by the first and the second equations of (114), i.e.,

$$\eta = -\frac{1}{2}\left[\frac{\partial\mathcal{K}(m,q)}{\partial q}\right]^{-1}\frac{\partial\mathcal{K}(m,q)}{\partial m}. \tag{147}$$

$\delta := \frac{\sqrt{\hat{q}}}{\hat{m}}$ is given by the the second and the third equations (114), i.e.,

$$\alpha\delta^2 = \left[\frac{\partial\mathcal{K}(m,q)}{\partial m}\right]^{-2}\mathcal{K}(m,q). \tag{148}$$

The special about the interpolation peak is that we need to expand $\mathcal{M}(\Sigma, m, q)$ to the second order, i.e.,

$$\mathcal{M}(\Sigma, m, q) = \frac{1}{2\Sigma}\mathcal{K}(m,q) + \frac{1}{2\Sigma^2}K_1(m,q), \tag{149}$$

where

$$K_1(m,q) := -\mathbb{E}_{z,z_0}\frac{1}{2}(h_0(z,z_0) - z)^T H(z,z_0)^\dagger(h_0(z,z_0) - z) \tag{150}$$

with

$$h_0(z,z_0) := \arg\inf_{h\in\arg\inf ||\sigma_1(z_0)-\sigma_2(\cdot)||^2} ||h - z||^2 \tag{151}$$

and

$$H(z, z_0) = \nabla_h^2 ||\sigma_1(z_0) - \sigma_2(h)||^2|_{h=h_0(z,z_0)}. \tag{152}$$

Then the first equation of (114) gives

$$\begin{aligned}
\Sigma\hat{\Sigma} &= 2\alpha\frac{\partial\mathcal{K}(m,q)}{\partial q} + \frac{2\alpha}{\Sigma}\frac{\partial K_1(m,q)}{\partial q} \\
&\approx 2\alpha\frac{\partial\mathcal{K}(m,q)}{\partial q} + \frac{8\alpha}{\eta\epsilon\delta}\frac{\partial K_1(m,q)}{\partial q},
\end{aligned} \tag{153}$$

where we use $\Sigma \approx \frac{\eta\epsilon\delta}{4}$ from the last equation of (114). Then combine it with the the last equation of (114), we obtain

$$4\alpha\delta^2\frac{\partial\mathcal{K}(m,q)}{\partial q} + \frac{16\alpha\delta}{\eta\epsilon}\frac{\partial K_1(m,q)}{\partial q} = \delta\partial_1 J(\delta,\epsilon) \approx \delta^2 - \frac{8}{3\pi}\epsilon\delta, \tag{154}$$

which implies

$$\begin{cases} \alpha = \frac{1}{4}\left[\frac{\partial\mathcal{K}(m,q)}{\partial q}\right]^{-1} \\ \frac{16\alpha}{\eta\epsilon}\frac{\partial K_1(m,q)}{\partial q} = -\frac{8}{3\pi}\epsilon. \end{cases} \tag{155}$$

Finally the generalization error is given by $\mathcal{I}(m,q)$, where $m,q$ are solved self-consistently from (145), (146), (147), (148), (155).

For the softmax activations, $M$ is linear in $q$, and thus the first equation of (155) gives a simple formula for the interpolation threshold

$$\alpha = \frac{1}{4}\left[\frac{1}{2}T(T+1) - 1\right]^{-1} \tag{156}$$

by using (113). Similarly, for invertible activations we have

$$\alpha = \frac{1}{2T(T+1)} \tag{157}$$

by using (112), which recovers the interpolation threshold $\frac{1}{4}$ in Erba et al. (2025) by choosing $T = 1$.

## J  DETAILS OF THE IMPLEMENTATION AND ADDITIONAL EXPERIMENTS

The analytical predictions in all plots are obtained by iterating the equations in Eq. (46) until convergence. The minimization in Eq. (13) is performed with the minimize package of Scipy, initializing in a Gaussian of variance $10^{-4}$ centered on $z_0$. The expectations in Eq. (46) are computed using Monte-Carlo integration with at least $10^4$ samples. Even though a single iteration of Eq. (46) takes typically less than one hour on a standard laptop, for convenience we used 60 nodes with 2 Intel Xeon 8360Y CPUs. For producing this paper we used approximately 60000 CPU hours including the initial exploration.

We offer some additional numerical explorations of the model in Eq. (1) that are complementary for the single-head tied-attention model in the main text.

In Fig. 6 we demonstrate a perfect match between GD and theory for various values of $\lambda$, which also suggests a large but finite interpolation peak at small regularization.

In Fig. 7 we evaluate the replicon condition (12). Let's define the replicon as

$$\text{Replicon} = 1 - 2\alpha\mathbb{E}_{z_0,z}\sum_{\substack{a,b,c,d=1 \\ a\leq b,\, c\leq d}}^{T}\left(\frac{\partial_{z_{ab}}p_{cd} - \delta_{ac}\delta_{bd}}{\Sigma}\right)^2\int\mu_{\sqrt{\hat{q}}/\hat{m}}(\mathrm{d}x)\,\mu_{\sqrt{\hat{q}}/\hat{m}}(\mathrm{d}y)\frac{(\xi(x)-\xi(y))^2}{\hat{\Sigma}^2(x-y)^2}. \tag{158}$$

In order for our results to hold we need the replicon to be positive.

In Fig. 8 left we display the thresholds for perfect recovery of the target in the limit of vanishing noise $\Delta \to 0$ and Marchenko-Pastur target. In Fig. 8 right we compare the prediction of the test error (14) with the error decomposition (126), which is an expansion around zero test error and thus qualitative in nature when applied to regions where the test error is far from zero. For the specific plot we used the following prescriptions:

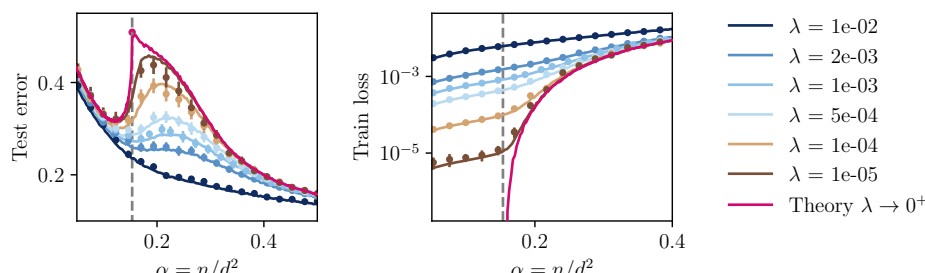

Figure 6: Test error (left) and training loss (right) as functions of the number of samples for decreasing regularization $\lambda$ ($\Delta = 0.5$, $T = 2$, $\beta = \beta_0 = 1$, $\kappa_0 = 0.5$, $\kappa = 1$). Solid lines: theoretical predictions; dots: Adam simulations at $d = 100$, averaged over 16 runs with 2000 samples in the test set. The gray dashed line marks the analytical interpolation threshold (Corollary B.2), i.e., the largest $\alpha$ for which the ERM estimator fits the training data as $\lambda \to 0^+$. As expected, the training loss vanishes before interpolation (vertical log scale), while the test error exhibits a non-symmetric interpolation peak, distinct from the usual cusp-like shape.

1. We approximate $\delta K'(\delta) \approx 0$, as it's of a smaller order in $d$ than the other terms for power-law targets;

2. We multiply the overfitting term of (126) by the factor $1 - K(\delta)/d$, to take into account the fact that the bulk will lose mass if a large number of spikes exit from it;

3. We take $\delta$, $\epsilon$, $\eta$ from the numerical solution of (46);

4. For the mismatch part of the error, we assume that $m = q = 1$ for $n < d^2/4$ instead of using the solution of (46), consistently with the theory (see Eq. (122)).

We remark that even though this procedure is qualitative, the agreement with the pure state evolution error is quite nice.

In Fig. 9 we compare the sequence-to-sequence and sequence-to-label formulations, evaluating test error via Eq. (3) (left) and training loss via Eq. (2) (right), for both noiseless ($\Delta = 0$) and noisy ($\Delta = 0.5$) settings at $T = 2$. The two variants are indistinguishable within error bars across the whole range of $\alpha = n/d^2$, confirming the asymptotic equivalence used in our proof sketch. The learning curves, indeed, exhibit the same phenomenology for the two variants of the model, seq2seq and seq2lab.

We next move to analyze the impact of the inverse temperature used inside the softmax activations in both the model and the target of Eq. (1).

In Fig. 10 we compare the cases in which the inverse temperature matched temperatures $\beta = \beta_0 \in \{0.5, 1, 2\}$ at fixed $T = 2$. Lower temperatures (larger smoothing of the row-wise softmax) systematically yield lower test error for a given $\alpha$, both without noise (left) and with label noise $\Delta = 0.5$ (right).

To conclude, we vary the number of tokens $T$ to investigate the impact of such parameter in the model's behavior. Finally, in Fig. 11 we vary the number of tokens $T \in \{2, 3, 5\}$ and plot the test error against the rescaled sample ratio $\bar{\alpha} = n/d^2\big(T(T+1)/2 - 1\big) = 2n/(T^2 + T - 2)d^2$, motivated by the effective number of scalar constraints in the attention matrix (cf. Appendix F). After this normalization, the learning curves for different $T$ largely align in both the noiseless and noisy cases (left/right), exhibiting the same qualitative dependence on the number of samples and confirming that $T$ primarily rescales the usable information rather than altering the underlying learning dynamics.

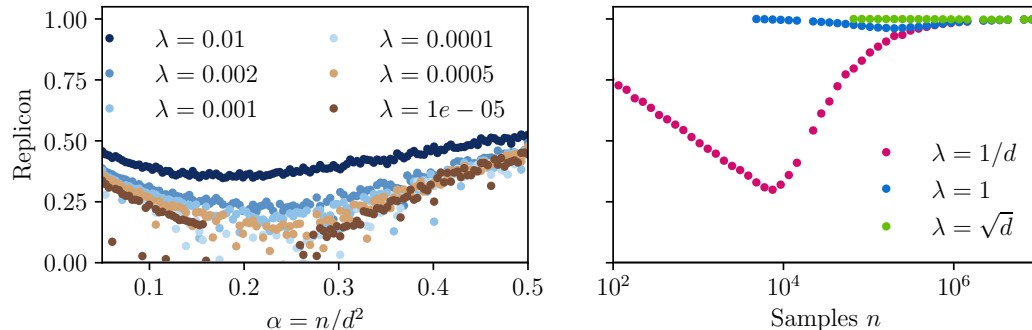

Figure 7: Replicon (158) condition for the same parameters as Figure 6 (left, including the data showed in Figure 1) and ad Figure 4. We see that in general the replicon is well bounded away from zero, meaning that the replicon condition (12) for Claim B.1 is satisfied. We remark that as $\lambda$ decreases, the replicon approaches lower and lower values around the interpolation threshold, where our numerical solver performs poorly. We conjecture that the condition is satisfied for all values of $\alpha$.

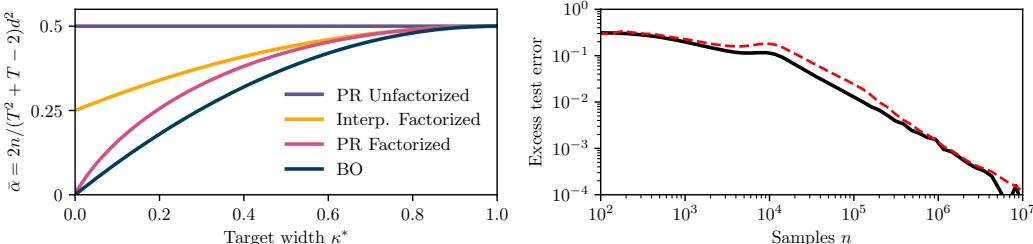

Figure 8: (Left) Analytical thresholds for vanishing regularization in the noiseless case $\Delta = 0$. We plot the Bayes-optimal perfect recovery from Boncoraglio et al. (2025) (lower-bound to the perfect recovery of any estimator), the perfect recovery of the attention model Eq. (2) from Corollary B.2, the interpolation threshold from Corollary B.2, and the perfect recovery of the non-factorized attention model Eq. (21). (Right) Comparison of the excess test error from Claim B.1 (in black) in tis non-asymptotic version as described in Section 4 and its equivalent computed using the error decomposition in Eq. (8). Parameters are $\gamma = 0.75$, $d = 200$, $\Delta = 0.5$, $T = 2$, $\lambda = 1/d$. The specific formula used is Eq. (126).

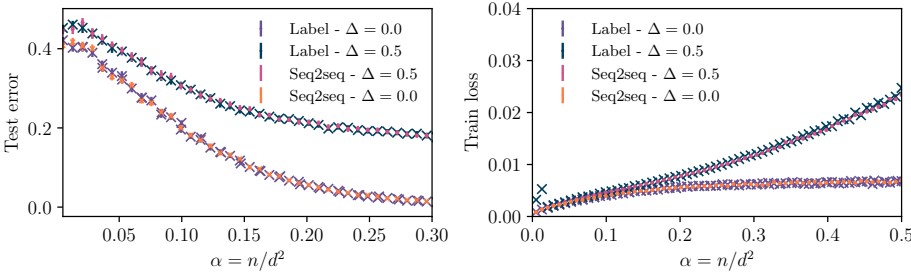

Figure 9: Comparison between the seq2seq model and the seq2lab model, both in Eq. (1). Both panels display the numerical simulations obtain from Adam with $\lambda = 0.01$, $\beta = \beta_0 = 1.0$, $T = 2$. We compare in both panels the noiseless $\Delta = 0.0$ and the noisy version of the model with $\Delta = 0.5$. We show in the (Left panel) the test error of the model computed from the seq2seq and the seq2lab formulas in Eq. (3). In the (Right panel) we show the train loss computed from the formulas in Eq. (2) with Adam. In all curves we average over 32 different realizations, with $d = 100$ and 2000 samples in the test set. Both variants achieve same test error and train loss, both in the noiseless and in the noisy version of the model.

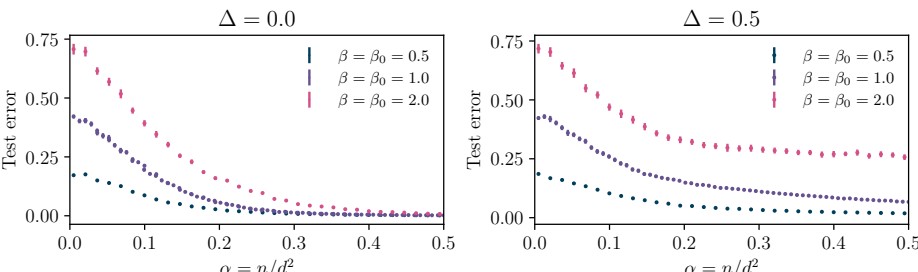

Figure 10: We show the test error for the model in Eq. (1) with respect to $\alpha = n/d^2$. We compute the test error through Adam for different values of the inverse temperatures $\beta = \beta_0$ of the softmax activation in the single-head tied-attention model. In all cases we consider the matched-temperature scenario between the target and the model. Both panels display the numerical simulations obtain from Adam with $\lambda = 0.01$, $\kappa_0 = 0.5$, $\kappa = 1$, $T = 2$. We compare in both panels the noiseless $\Delta = 0.0$ (Left panel) and the noisy version of the model with $\Delta = 0.5$ (Right panel) for three values of the inverse temperatures $\beta = \beta_0 = 0.5, 1, 2.0$. In all curves we average over 32 different realizations, with $d = 100$ and 2000 samples in the test set.

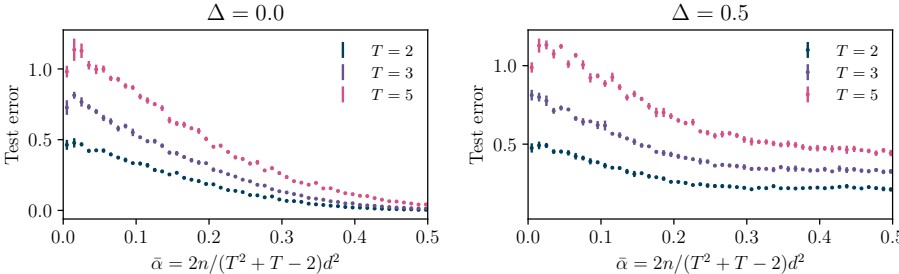

Figure 11: We show the test error computed through Adam for the model in Eq. (1) for a different number of tokens $T \geq 2$. In the x-axis we show the rescaled sample complexity $\bar{\alpha} = n/d^2 \left( \frac{T(T+1)}{2} - 1 \right) = 2n/(T^2 + T - 2)d^2$. Both panels display the numerical simulations obtain from Adam with $\lambda = 0.001$, $\kappa_0 = 0.5$, $\kappa = 1$, $\beta = \beta_0 = 1.0$ and different number of tokens $T = 2, 3, 5$. We compare in both panels the noiseless $\Delta = 0.0$ (Left panel) and the noisy version of the model with $\Delta = 0.5$. In all curves we average over 32 different realizations, with $d = 100$ and 2000 samples in the test set.

