# OpenReview forum: "Single-Head Attention in High Dimensions: A Theory of Generalization, Weights Spectra, and Scaling Laws"
_ICLR.cc/2026/Workshop/Sci4DL — Sci4DL 2026_

### Official Review · Reviewer_bUmP · 2026-02-12

**Fit:** 3
**Significance:** 2
**Confidence:** 1

**Summary:**

In this paper, the authors analyze the properties of solutions for learning a (simplified) single-head attention layer. They are able to derive analytical expressions for both the generalization error and the spectral distribution of the learned attention matrix based on a regularized ERM rule. The theory can quantitatively reproduce the low-rank properties of attention layers in transformers.

**Strengths:**

The authors try to address a very important question about the inductive bias in transformer models using a theory-first approach, combined with simulation results. The analytical results accurately match the simulations of both the generalization error and the low-rank spectral distribution, and demonstrate very cleanly the effect of factorized representations.

**Suggestions:**

I personally find the paper very difficult to follow (partially due to the four-page limit, which requires going back and forth between the main paper and the supplement). In particular, I have difficulty connecting the theoretical results to the conceptual claims the authors are trying to make.

Is the main conclusion that the low-rank structure arises from the regularization on the W matrix, which effectively becomes a nuclear norm regularization in the theory? If so, is the central claim that weight decay is crucial to the resulting structure in the weight matrix of attention layers? What additional insight does the theory provide into the effect/strength of weight decay, or other aspect of how the properties of the data/target function interact with the learned solution?

In general, I had trouble telling apart the technical details of the theory from the broader conceptual implications. For example, the authors introduce quite a few quantities in Eq. 9, but I have difficulty understanding which of them are most important and what the key axes are in terms of the properties of the learned solution.

Lastly, the authors choose a simplified model with the same key and query matrix and an identity value matrix. While I understand the necessity of simplification for theoretical analysis, I would appreciate some discussion of why this particular simplification was made and what aspects the theory might be missing as a result.

---

### Official Review · Reviewer_4xmT · 2026-02-27

**Fit:** 3
**Significance:** 2
**Confidence:** 2

**Summary:**

This paper theoretically analyzes the generalization properties of single-head attention along with the spectra of weight matrices on a seq2seq task and attention map prediction task, in a high-dimensional, realizable setting.

**Strengths:**

Rigorous analysis of generalization properties for attention, with both theory and empirical evidence. Connecting the generalization properties to spectral structure is very interesting.

**Suggestions:**

I’m curious how the results would look in more realistic training setups, e.g., if the target function is not realizable, or if $n$ scales faster than $O(d^2)$. I understand this is not the focus of this paper, however a discussion of these aspects would nevertheless be informative to a broader audience.

---

### Meta-Review · Area_Chair_pe2K · 2026-03-01

**Recommendation:** Accept

**Metareview:**

The paper addresses an important research question and makes significant contributions that are worth sharing at the workshop. However, clarity of the presentation can be improved, and I suggest the authors do so in the camera-ready version.

---

### Decision · Program_Chairs · 2026-03-02

Accept